# BLACK-BOX ATTACK ROBUSTNESS WITH MODEL DIVERSITY AND RANDOMIZATION

## ABSTRACT

Query-based black-box attack algorithms compute imperceptible adversarial *perturbations* to misguide learned models, relying *only* on model outputs. The success of these algorithms poses a significant risk, especially for Machine Learning as a Service (MLaaS) providers. We explore a new approach to obfuscate information from an attacker. To craft an adversarial example, attacks *exploit* inter-query relationships to optimize a perturbation. We investigate if the relationship can be *obfuscated* by randomizing model *parameters* in contrast to sate-of-the-art approaches' reliance on random noise. Effectively, randomization violates the attacker's assumption of fixed *model parameters* between queries to extract information. *What is unclear is, if model randomization can lead to sufficient obfuscation or how best to build such a method.* We seek answers to these questions. Our theoretical analysis proves the approach consistently increases robustness. Extensive experiments across 7 state-of-the-art attacks and all major perturbation norms ($l_\infty$, $l_2$, $l_0$), including adaptive variants, confirm its effectiveness. Importantly, our findings reveal a new avenue for investigating robust methods against black-box attacks, offering theoretical understandings and a practical implementation pathway.

## 1 INTRODUCTION

In white-box settings, algorithms exploit full model access to launch powerful attacks like Projected Gradient Descent. These attacks craft and apply imperceptible perturbations to inputs to mislead or hijack the decision of deep learning models (Szegedy et al., 2014; Papernot et al., 2017; Carlini & Wagner, 2017; Madry et al., 2018; Athalye et al., 2018). However, practical deployments of models, with growing numbers of machine learning as a service (MLaaS) offerings, restrict access to model internals. Consequently, attackers are limited to query-response based black-box attacks exploiting only model outputs. These attacks pose a practical threat, as demonstrated in real-world systems Ilyas et al. (2018); Guo et al. (2019); Vo et al. (2024).

Query-based attack algorithms succeed because they can extract information to move in a direction towards an adversarial example, simply using model response differences to inputs with small changes. Naturally, a defender's objective is to prevent the extraction of *useful* information from responses.

**Our Study.** We investigate *if* the defense objective can be achieved by randomizing *parameters* of a model in contrast to sate-of-the-art approaches' reliance on random noise.

> Crucially, adversarial attacks depend on successive queries and their responses to gradually move a source class towards a target (non-source) class' decision boundary, as seen in Figure 1 with model $\theta_1$. *The key insight behind our defense is that this process relies on fixed model parameters—any change between queries disrupts the relationship needed to optimize the perturbation.*

So, our proposal to randomize model parameters should obfuscate the relationship between successive queries and responses to confuse the iterative optimization process. To achieve obfuscation through model parameter randomization, we investigate sampling models from a set of *diverse* models to respond to each query as illustrated in the *last* tile of Figure 1. Then, to minimize potential impacts of our defense strategy on performance, we investigate learning a set of *well-performing* models.

Our theoretical analysis shows the diversity of responses from randomly sampled models can introduce sufficient uncertainty to degrade gradient estimates or misdirect random search attempts

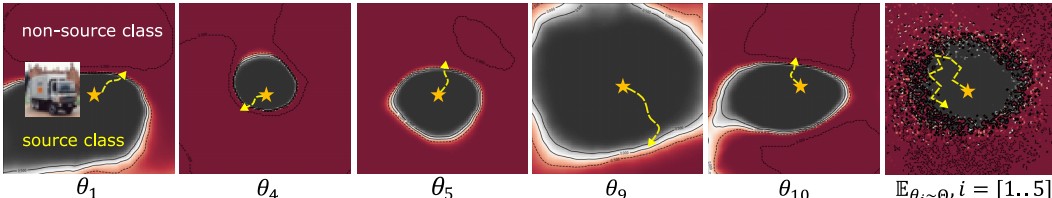

$\theta_1$ $\qquad\qquad$ $\theta_4$ $\qquad\qquad$ $\theta_5$ $\qquad\qquad$ $\theta_9$ $\qquad\qquad$ $\theta_{10}$ $\qquad\qquad$ $\mathbb{E}_{\theta_i \sim \Theta}, i = [1..5]$

Figure 1: Visualization of decision boundaries for *randomly* sampled 5 of the 10 well-performing, diverse models ($\theta_1, \theta_2, ..., \theta_{10}$) around a clean input from the *source class* Truck in CIFAR-10. Using only model outputs, a query-based attacker can estimate gradients or search for a path toward the decision boundary to craft adversarial examples against models $\theta_1, \theta_4, \theta_5, \theta_9$ & $\theta_{10}$. However, when responses to queries are returned from *randomized models*, it is significantly harder to estimate gradients or search for a path toward the decision boundary as we show in the *last tile*. Here, each response to a query is generated from five randomly sampled models and the attack is misguided. We investigate *if* model randomization can lead to sufficient obfuscation to confuse attacks and *how* best to build such a method.

from attacks. Consequently, building adversarial examples with *score-based* or *decision-based* attack algorithms are made significantly more difficult. Our key contributions can be summarized as follows:

- We investigate the effectiveness of randomized sampling of diverse and well-trained models as a defense with a theoretical analysis.

- We propose promoting *model diversity* and because we also want high utility for random model parameter selections, we introduce a *new* learning objective to diversity promotion to ensure models are *well-performing*. The defense framework we dubb Disco[1] is flexible to incorporate other model diversity promotion methods or even existing random noise techniques.

- Extensive evaluations with both score-based and decision-based attacks as well as all *three* perturbation objectives ($l_\infty, l_2, l_0$), including testing with CLIP on ImageNet, validate our *theoretical analysis* and show our method can enhance robustness against query-based attacks.

## 2 BACKGROUND AND RELATED STUDIES

**Query-based Black-Box Attack Primer.** In contrast to white-box attacks, black-box attackers do not have access to a victim model. An approach is transfer-based black-box attacks that craft adversarial examples from a surrogate model and transfer to a victim model (Papernot et al., 2017; Chen & Liu, 2023). But, transfer-based attacks' success varies significantly due to factors such as model hyperparameters, training conditions and constraints in generating adversarial samples (Chen et al., 2017) and similarity between the surrogates and target models (Suya et al., 2024). In this paper, we focus on defending against query-based black-box adversarial attacks. These attacks submit an input to obtain a response from a model. When the response is a confidence score, the attacks operate in a *score-based* threat model; when the response is a label, the attacks operate in a *decision-based* threat model. Two primary approaches to query-based attacks are:

- Gradient estimation methods (Liu et al., 2018a; Ilyas et al., 2018; Cheng et al., 2020; Chen et al., 2020a) estimate the model's gradient with respect to an image $x$ by exploring images surrounding $x$ with queries to assess the model's gradient.
- Gradient-free methods (Ru et al., 2020; Andriushchenko et al., 2020; Croce et al., 2022; Vo et al., 2022a; Cheng et al., 2024) introduce small random modifications to an image $x$ and observe query response to assess the perturbation's goodness without gradient information.

In our work, we use *both* score-based and decision-based attacks. In contrast to past studies, we evaluate attack algorithms covering *three* perturbation objectives ($l_\infty, l_2, l_0$).

**General Defenses.** Adversarial training, as a more general method, can be used to defend against adversarial attacks (Tramèr et al., 2018; Zhang et al., 2020; 2022). Similarly, training with noise (Cohen et al., 2019; Salman et al., 2019) makes models robust against adversarial inputs. But, these approaches diminish model performance (Zhang et al., 2019; Shafahi et al., 2019; Yang et al., 2020).

---

[1]Diversity Induced Stochastic Obfuscation. Code will be at https://github.com/disco-defense/.

**Defenses Against Query-Based Attacks.** Methods exploiting the anomalous nature of queries attempt to detect attacks (Chen et al., 2020b; Pang et al., 2020; Li et al., 2022). An alterative school of thought studies: i) adding noise to inputs Cao & Gong (2017); Qin et al. (2021); ii) injecting noise to model's parameters, activation or adding noise layers (Liu et al., 2018b; He et al., 2019) or iii) adding noise to features (Nguyen et al., 2024) to misdirect the query-based search. Primarily, we evaluate the following recent defenses:

- *Randomize Noise Defense (RND).* Qin et al. (2021) introduced adding random noise to the input and theoretically analyzed the effectiveness against query-based attacks. Byun et al. (2022) also proposed a similar method dubbed SND. Thus, our comparison with RND will extend to both.
- *Randomize Features.* Nguyen et al. (2024) proposed adding random noise in feature space.

In contrast to studies employing random noise, we investigate randomization in the function space—effectively the parameter space—to *distort* information available in responses to *deceive* attackers.

**Adversarial Robustness with Ensemble Diversity.** Prior studies (Kariyappa & Qureshi, 2019; Pang et al., 2019; Doan et al., 2022) explored diversity of ensembles to improve adversarial robustness in *white-box* settings but at the cost of sacrificing clean accuracy (Tsipras et al., 2019; Qin et al., 2021). In contrast, we study ensemble diversity for constructing well-performing, model parameter alternatives. Notably, a number of diversity promotion methods exist in the literature—we elaborate further in **Appendix H.1**. In this paper, we evaluate with: i) Deep Ensembles (Ensemble) (Lakshminarayanan et al., 2017; Fort et al., 2020; Wen et al., 2020); ii) DivDis (Lee et al., 2023); iii) DivReg (Teney et al., 2022) and iv) together with a learning objective we propose for Stein Variational Gradient Descent (SVGD) method to learn diverse and *well-performing* models.

## 3 PROPOSED METHOD

In this section, we formally describe the threat as a problem description, explain our thinking behind our approach for confusing attackers with model randomization, and then provide a theoretical analysis of the convergence of attack algorithms under our our defense method.

### 3.1 PROBLEM FORMULATION

**Score-based Settings.** Given a benign input $\boldsymbol{x} \in \mathbb{R}^d$ and ground truth label $y$, let $f(\boldsymbol{x}, \boldsymbol{\theta})$ denote a victim model with logit score outputs. In untargeted settings, the focus in defense domains, the goal of an adversary is to search for an adversarial example $\tilde{\boldsymbol{x}} \in \mathbb{R}^d$ such that $\arg\max_{\tilde{y}} p(\tilde{y} \mid \tilde{\boldsymbol{x}}) \neq y$ and $\|\boldsymbol{x} - \tilde{\boldsymbol{x}}\|_p \leq B$, where $p(\tilde{y} \mid \tilde{\boldsymbol{x}}) = \text{softmax}\left[f(\tilde{\boldsymbol{x}}; \boldsymbol{\theta})\right]$, $\|.\|_p$ denotes $l_p$ norm and $B$ represents the perturbation budget. Two main approaches to score-based attacks are:

- *Gradient Estimation with Finite Difference Method.* An adversary estimates the gradient based on the average difference between pairs of model's output scores with $\boldsymbol{u}_i \sim \mathcal{N}(0, \boldsymbol{I})$ as follows:

$$\tilde{g}_s(\tilde{\boldsymbol{x}}) = \frac{1}{n} \sum_{i=1}^{n} \frac{f(\tilde{\boldsymbol{x}} + \epsilon \boldsymbol{u}_i; \boldsymbol{\theta}) - f(\tilde{\boldsymbol{x}}; \boldsymbol{\theta})}{\epsilon} \boldsymbol{u}_i. \tag{1}$$

- *Gradient-free methods.* An adversary can employ random search algorithms that determine attack direction based on the $f(\tilde{\boldsymbol{x}} + \epsilon \boldsymbol{u}; \boldsymbol{\theta}) - f(\tilde{\boldsymbol{x}}; \boldsymbol{\theta})$. An attack direction is successful if $f(\tilde{\boldsymbol{x}} + \epsilon \boldsymbol{u}; \boldsymbol{\theta}) - f(\tilde{\boldsymbol{x}}; \boldsymbol{\theta}) < 0$.

**Decision-based Settings.** The adversarial objective (untargeted attacks) is to minimize distance $D(\boldsymbol{x}, \tilde{\boldsymbol{x}}) = \|\boldsymbol{x} - \tilde{\boldsymbol{x}}\|_p$ such that $\arg\max_{\tilde{y}} p(\tilde{y} \mid \tilde{\boldsymbol{x}}) \neq y$. Similar to score-based settings, to achieve this objective, an adversary can employ gradient estimation or gradient-free methods. For gradient estimation methods, the gradient can be estimated with $\boldsymbol{u}_i \sim \mathcal{N}(0, \boldsymbol{I})$ as follows:

$$\tilde{g}_d(\tilde{\boldsymbol{x}}) = \frac{1}{n} \sum_{i=1}^{n} \frac{D(\boldsymbol{x} + \epsilon \boldsymbol{u}_i, \tilde{\boldsymbol{x}}) - D(\boldsymbol{x}, \tilde{\boldsymbol{x}})}{\epsilon} \boldsymbol{u}_i. \tag{2}$$

### 3.2 MODEL RANDOMIZATION APPROACH FOR OBFUSCATING INFORMATION

Query-based black-box attack algorithms depend on multiple queries and model responses to estimate a gradient or a search direction as we describe in the problem formulation (Section 3). Our goal is to

obfuscate the relationship between query-response pairs. For example, consider the query-response—$\tilde{\boldsymbol{x}} + \epsilon \boldsymbol{u}$ and $f(\tilde{\boldsymbol{x}} + \epsilon \boldsymbol{u}; \boldsymbol{\theta})$—with query-response—$\tilde{\boldsymbol{x}}$ and $f(\tilde{\boldsymbol{x}}; \boldsymbol{\theta})$—for a score-based attack described in equation 1. Now, we can simply remove the attacker's assumption of fixed model parameters $\theta$ and randomize this for each query-response so the estimated gradient is conditional on some parameter $\theta$ unknown to an attacker. This simple change violates the key assumption of fixed model parameters across queries to extract a reliable information for an attack.

By employing a *different* function (a learned model $\theta$) to process a query input and generate a response, we can expect to hide the relationship between query-response pairs. Because the attacks rely on this relationship, model randomization should lead to sufficient uncertainty to misguide the search direction towards an adversarial example. Consequently, we hypothesize that:

- **Hypothesis 1**. *Randomly selecting a model from a set for a query response can obfuscate the relationship between successive pairs of queries and responses.*
- **Hypothesis 2.** *Enhancing model parameter diversity enhances obfuscation since diverse parameters should increase variations in model outputs to degrade the information extracted from query-response pairs.*

In the following, we investigate these hypotheses to establish our approach.

## 3.3 FORMULATING RANDOM MODEL SELECTION

Following our *first* hypothesis, we expect feedback from randomly selected models to misdirect gradient estimation and search direction algorithms. For generality, in iteration $i$, we consider randomly selecting a subset of models rather than a single model. A single models is a special case of the selected set having only one model. Then, given a set of models $\mathcal{F} = \{f(\cdot, \boldsymbol{\theta}_1), f(\cdot, \boldsymbol{\theta}_2), \ldots, f(\cdot, \boldsymbol{\theta}_K)\}$ and each model $f(\cdot, \boldsymbol{\theta}_k) \in \mathcal{F}$ with parameters $\boldsymbol{\theta}_k$, where $K$ is the total number of models. The prediction of a subset of $N$ models sampled from $K$ models can be formulated as follows:

$$y^* = \arg\max_y p(y \mid \boldsymbol{x}), \tag{3}$$

where $p(y \mid \boldsymbol{x}) = \text{softmax}[q(\boldsymbol{x}; \boldsymbol{\pi})]$, function $q(\boldsymbol{x}; \boldsymbol{\pi}) = \frac{1}{N} \sum_{k=1}^{N} \pi_k f(\boldsymbol{x}, \boldsymbol{\theta}_k)$, and $\boldsymbol{\pi} \sim \mathcal{B}(\mu_1, \ldots, \mu_K)$ denoting $K$ dimensional vector sampled from $K$ independent Bernoulli distributions. $\mu_k$ is the mean of a Bernoulli distribution denoting the expected number of times a model is selected.

## 3.4 THEORETICAL ANALYSIS AGAINST GRADIENT ESTIMATION ATTACKS

Consider a constant $\epsilon > 0$, $\boldsymbol{u} \sim \mathcal{N}(0, \boldsymbol{I})$, $\boldsymbol{u} \in \mathbb{R}^d$ and $\boldsymbol{x} \in \mathbb{R}^d$, with the output logits of all models expressed as $F(\boldsymbol{x}) = \frac{1}{K} \sum_{k=1}^{K} f_k(\boldsymbol{x}; \boldsymbol{\theta}_k)$ and with a slight misuse of notation, the gradient of such a totality of models can be estimated as follows:

$$\hat{G}(\boldsymbol{x}) = \mathbb{E}_{\boldsymbol{u}} \left[ \frac{F(\boldsymbol{x} + \epsilon \boldsymbol{u}) - F(\boldsymbol{x})}{\epsilon} \boldsymbol{u} \right]; \quad \hat{G}(\boldsymbol{x}) \in \mathbb{R}^d. \tag{4}$$

Under our model randomization approach described in Equation 3, the gradient estimator for a pair of input query samples is:

$$g(\boldsymbol{x}) = \frac{q(\boldsymbol{x} + \epsilon \boldsymbol{u}; \boldsymbol{\pi}^{(2)}) - q(\boldsymbol{x}; \boldsymbol{\pi}^{(1)})}{\epsilon} \boldsymbol{u}; \quad g(\boldsymbol{x}) \in \mathbb{R}^d. \tag{5}$$

Then, the approximation of the gradient with $n$ different pairs of samples using the finite difference method is formulated as follows:

$$\bar{g}(\boldsymbol{x}) = \frac{1}{n} \sum_{i=1}^{n} \frac{q(\boldsymbol{x} + \epsilon \boldsymbol{u}_i; \boldsymbol{\pi}^{(2i)}) - q(\boldsymbol{x}; \boldsymbol{\pi}^{(2i-1)})}{\epsilon} \boldsymbol{u}_i; \tag{6}$$

where $\bar{g}(\boldsymbol{x}) \in \mathbb{R}^d$, defenders generate $\boldsymbol{\pi}^{(2i)} \boldsymbol{\pi}^{(2i-1)} \sim \mathcal{B}(\mu_1, \ldots, \mu_K)$ while attackers generate $\boldsymbol{u}_i \sim \mathcal{N}(0, \boldsymbol{I})$.

**Proposition 3.1.** *Consider an input $\boldsymbol{x}$ where each element of the gradient $g(\boldsymbol{x})$ estimated at iteration $i$ given in Equation 5 is bounded by $a_i^j \leq g(\boldsymbol{x})^j \leq b_i^j$, where $\boldsymbol{a}_i, \boldsymbol{b}_i \in \mathbb{R}^d$, and the average gradient*

*estimator is $\bar{g}(\boldsymbol{x})$ as defined in Equation 6. Then, the number of samples $n$ needed such that for every element of $\bar{g}(\boldsymbol{x}) - \hat{G}(\boldsymbol{x})$ is within an error margin $\Delta$ with confidence $1 - \delta$ is at least:*

$$n \geq \sqrt{\frac{\log(\frac{2d}{\delta}) \sum_{i=1}^{n} \left[\max_j (b_i^j - a_i^j)\right]^2}{2\Delta^2}} \tag{7}$$

*Proof.* We defer the proof to Appendix A ∎

> Proposition 3.1 states our proposal effectively fortifies against query-based attacks where the cost of the attack, the queries $n$ required to drive $\bar{g}$ closer to $\hat{G}$, is made large to thwart attacks. Moreover, this cost depends on the gradient estimator's bounds, $\boldsymbol{a}_i$ and $\boldsymbol{b}_i$; interestingly, this can be made large when the underlying set of models are able to generate highly diverse outputs to given pairs of inputs.

Importantly, proposition 3.1 still holds true for *adaptive attacks* based on Expectation Over Transformation (Athalye et al., 2017) as detailed in Appendix A. The results in Sections 4.1 and 4.2 confirm our observation about the effectiveness and robustness of our defense against *adaptive attacks*.

### 3.5 THEORETICAL ANALYSIS AGAINST GRADIENT-FREE ATTACKS

Consider a constant $\epsilon > 0$ and $\boldsymbol{u} \sim \mathcal{N}(0, \boldsymbol{I})$. Then, the search direction of gradient-free methods against the ensemble of all of the models relies on the sign of $\hat{H}(\boldsymbol{x}, \boldsymbol{u})$ expressed as $\text{sign}(F(\boldsymbol{x} + \epsilon\boldsymbol{u}) - F(\boldsymbol{x}))$, while the search direction of gradient-free methods against randomly selected subsets of models depends on the sign of $\tilde{H}(\boldsymbol{x}, \boldsymbol{u})$ formulated as $\text{sign}(q(\boldsymbol{x} + \epsilon\boldsymbol{u}; \boldsymbol{\pi}^{(i)}) - q(\boldsymbol{x}; \boldsymbol{\pi}^{(j)}))$. As different signs between $\hat{H}(\boldsymbol{x}, \boldsymbol{u})$ and $\tilde{H}(\boldsymbol{x}, \boldsymbol{u})$ or in other words, $\frac{\tilde{H}(\boldsymbol{x}, \boldsymbol{u})}{\hat{H}(\boldsymbol{x}, \boldsymbol{u})} < 0$, represents the mismatch between the attack directions against a random subset of models versus that generated using the entire set of models, $P\left(\frac{\tilde{H}(\boldsymbol{x}, \boldsymbol{u})}{\hat{H}(\boldsymbol{x}, \boldsymbol{u})} < 0\right)$ represents the probability of misleading attack directions.

**Proposition 3.2.** *If we define $\gamma_{i,j} = q(\boldsymbol{x}; \boldsymbol{\pi}^{(i)}) - q(\boldsymbol{x}; \boldsymbol{\pi}^{(j)})$ and $\zeta_i = \nabla q(\boldsymbol{x}; \boldsymbol{\pi}^{(i)}) - \frac{1}{K}\sum_{k=1}^{K} \nabla f_k(\boldsymbol{x}; \boldsymbol{\theta}_k)$, then the probability of misleading attack directions is bounded by:*

$$P\left(\frac{\tilde{H}(\boldsymbol{x}, \boldsymbol{u})}{\hat{H}(\boldsymbol{x}, \boldsymbol{u})} < 0\right) \leq \frac{\sqrt{2\mathbb{E}_\pi\left[\gamma_{i,j}^2 + (\epsilon\boldsymbol{u}\zeta_i)^2\right]}}{|\hat{H}(\boldsymbol{x}, \boldsymbol{u})|} \tag{8}$$

*Proof.* We defer the proof to Appendix B ∎

> Proposition 3.2 states that the probability of misleading a search-based attack is low if the model output diversity is low. Intuitively, the random selection of diverse models can result in diverse outputs; alternatively, $q(\boldsymbol{x}; \boldsymbol{\pi}^{(i)}) - q(\boldsymbol{x}; \boldsymbol{\pi}^{(j)})$ is positively correlated with the output diversity.

### 3.6 FORMULATING A METHOD TO ACHIEVE MODEL OUTPUT DIVERSITY

Our theoretical analysis of model randomization confirms that promoting model output diversity improves robustness to query-based black-box attacks. To achieve output diversity, we investigate our *second* hypothesis by considering methods to learn *diverse model parameters* to enhance model output diversity. In general, we can train an ensemble of models such that their predictions are consistent while their responses *i.e.* output scores are diverse. Formally, the training objective of such a framework can be formulated as follows:

$$\min_{\boldsymbol{\Theta}} \quad \mathbb{E}_{(\boldsymbol{x}, y) \sim \mathcal{D}} \mathcal{L}(\boldsymbol{x}, y), \text{ s.t. } \Omega(\mathcal{F}), \tag{9}$$

where $\mathcal{D}$ denotes a training set, $\Omega$ is a set of constraints on the set of functions $\mathcal{F} = \{f(\cdot, \boldsymbol{\theta}_1), \ldots, f(\cdot, \boldsymbol{\theta}_K)\}$ to ensure diversity is optimized over their parameters $\boldsymbol{\Theta} = \{\boldsymbol{\theta}_1, \ldots, \boldsymbol{\theta}_K\}$, $\mathcal{L}(\boldsymbol{x}, y) = \ell(F(\boldsymbol{x}), y)$ and $F(\boldsymbol{x}) = \frac{1}{K}\sum_{k=1}^{K} f_k(\boldsymbol{x}; \boldsymbol{\theta}_k)$ is a cross-entropy loss. For our defense, there are two pertinent questions that have to be answered in formulation of Equation 9:

**Question 1:** What constraints encourage the diversity of models leading to high output diversity?

**Question 2:** Since we desire randomly selected models to be *well-performing* to minimize the defense's impact on performance, how can we reconcile the asymmetry between promoting individual model performance whilst ensuring models diversity without model collapse?

### 3.6.1 PARAMETER DIVERSITY APPROACH FOR ACHIEVING MODEL OUTPUT DIVERSITY

We adopt a training framework incorporating a Bayesian formulation of deep learning with Stein Variational Gradient Descent (SVGD) method (Liu & Wang, 2016; Wang & Liu, 2019) to construct a diverse set of parameters. This framework enables learning a posterior distribution of parameters and parameters sampled from that posterior distribution can result in diverse representations Doan et al. (2022), leading to model parameter diversity and output variance without compromising accuracy. *Essentially, the SVGD method push model parameters apart directly and provide an effective solution to model diversity in* **Question 1**.

In this approach, a neural network $f(\mathbf{x}, \boldsymbol{\theta})$ with parameters $\boldsymbol{\theta}$ are considered random variables. Then Bayesian deep learning begins with a prior $p(\boldsymbol{\theta})$ and a likelihood function $p(\mathcal{D}|\boldsymbol{\theta})$ that assesses how well the network with weights $\boldsymbol{\theta}$ fits the data $\mathcal{D}$. The Bayesian inference integrates the likelihood and the prior using Bayes' theorem to derive a *posterior* distribution, $p(\boldsymbol{\theta}|\mathcal{D})$, over the space of weights, given by $p(\boldsymbol{\theta}|\mathcal{D}) = \frac{p(\mathcal{D}|\boldsymbol{\theta})p(\boldsymbol{\theta})}{p(\mathcal{D})}$. The exact solution for the posterior is often impractical, due to the complexity of deep neural networks and the high-dimensional integral of the denominator, even for networks of moderate size. The true Bayesian posterior distribution is complicated and challenging to accurately sample from. Therefore, Liu & Wang (2016) proposed SVGD as a general-purpose variational inference algorithm. Formally, learning diverse parameters is formulated as follows:

$$\boldsymbol{\theta}_i = \boldsymbol{\theta}_i - \epsilon\phi^*(\boldsymbol{\theta}_i)$$

$$\phi^*(\boldsymbol{\theta}_i) = \sum_{k=1}^{K}\Big[\kappa(\boldsymbol{\theta}_k, \boldsymbol{\theta}_i)\nabla_{\boldsymbol{\theta}_i}\mathcal{L}(\boldsymbol{x}, y) - \gamma\nabla_{\boldsymbol{\theta}_i}\kappa(\boldsymbol{\theta}_k, \boldsymbol{\theta}_i)\Big]. \tag{10}$$

where $\boldsymbol{\theta}_k$ denotes the weights of the $k$-th model, $\kappa(\cdot, \cdot)$ is a kernel function that encourages model diversity, and $\gamma$ is a hyperparameter to control the trade-off between models' diversity, $\epsilon$ is the learning rate and $\mathcal{L}(\boldsymbol{x}, y)$ from equation 9. Notably, SVGD method was first employed to improve adversarial robustness in white-box settings (Doan et al., 2022) with adversarial training. We avoid adversarial training due to the resulting clean accuracy drop. Importantly, the method proposed by Doan et al. (2022) does not consider the problem pertinent to our defence posed in **Question 2**.

### 3.6.2 NEW TRAINING OBJECTIVE TO ACHIEVE WELL-PERFORMING MODELS

We observe the training objective in Equation 9 is unable to address the problem posed in **Question 2** as shown in Appendix D. Simply, a naive adoption of the Bayesian training framework with SVGD does not yield individual models that perform well, despite the average performance of all models for a task being high. To address this problem, we encourage individual model-learning while training a set of diverse models. Formally, a new joint training objective with a new loss incorporating a *sample loss*, $\ell(f(\boldsymbol{x}; \boldsymbol{\theta}_i), y)$, $\boldsymbol{\theta}_i \sim \Theta$, with a given training set $\mathcal{D}$ is formulated as follows:

$$\min_{\Theta}\mathbb{E}_{\mathcal{B}\sim\mathcal{D},\,\boldsymbol{\theta}_i\sim\Theta}\Big[\mathbb{E}_{(\boldsymbol{x},y)\sim\mathcal{B}}\mathcal{L}(\boldsymbol{x}, y; \boldsymbol{\theta}_i)\Big], \tag{11}$$

where $\mathcal{L}(\boldsymbol{x}, y; \boldsymbol{\theta}_i) = \ell\Big(\frac{1}{K}\sum_{k=1}^{K}f(\boldsymbol{x}; \boldsymbol{\theta}_k), y\Big) + \ell\Big(f(\boldsymbol{x}; \boldsymbol{\theta}_i), y\Big)$. Notably, in this training framework, we aim to train all models simultaneously, and for each batch of data $\mathcal{B}$, we uniformly select $\boldsymbol{\theta}_i$ from $\Theta$ at random with replacement. The loss $\mathcal{L}(\boldsymbol{x}, y)$ in equation 10 is then replaced with $\mathcal{L}(\boldsymbol{x}, y; \boldsymbol{\theta}_i)$.

## 4 EXPERIMENTS AND EVALUATIONS

**Datasets & Models.** We use four different datasets `MNIST` (Lecun et al., 1998), `CIFAR-10` (Krizhevsky et al.), `STL-10` (Coates et al., 2011) and `ImageNet` (Deng et al., 2009). We use the network in (Cheng et al., 2020) for `MNIST`, VGG-16 (Liu & Deng, 2015) for `CIFAR-10` and ResNet18 (He et al., 2016) for `STL-10`, then OpenCLIP Radford et al. (2021) for `ImageNet`.

As discussed in Section 3.3, more diverse models can improve resilience to attack algorithms. Given computational constraints and the complexity of different datasets (*i.e.* high dimension), we train a larger number of models (40) for `MNIST` and a lower number of models (10) for high-resolution `CIFAR-10` and `STL-10`, 5 for `ImageNet`. Notably, with the ability to relatively quickly learn with a large number of models on `MNIST`, we conduct extensive studies using `MNIST`.

**Attacks.** We use both *score* and *decision*-based attacks across three perturbation objectives ($l_\infty$, $l_2$, $l_0$). We emphasize score-based attacks as state-of-the-art methods succeed with smaller query budgets. For *score-based* settings under $l_2$, $l_\infty$ and $l_0$ perturbation objectives, we attack with SQUAREATTACK Croce et al. (2022), NESATTACK (Ilyas et al., 2018), SIGNHUNTER (Al-Dujaili & O'Reilly, 2020) and SPARSERS (Andriushchenko et al., 2020). For *decision-based* settings we use HOPSKIPJUMP (Cheng et al., 2019) ($l_2$) and SPAEVO (Vo et al., 2022a) ($l_0$).

**Defenses.** We compare ours with: i) randomized input, ii) RND (Qin et al., 2021), and iii) randomized feature, RF (Nguyen et al., 2024), defenses for query-based attacks. Notably, comparing the empirical robustness of all adversarial defenses is beyond the scope of this paper. Our aim is to theoretically and empirically examine the effectiveness of our defense. Nevertheless, for completeness, we compare with 4 additional defenses: **iv)** adversarial training (AT) (Wang et al., 2023) in the **Appendix** M; **v)** AAA (Chen et al., 2022) defending against *only* score-based attacks in **Appendix** N; **vi)** RBC input randomization defense (Cao & Gong, 2017) in **Appendix** P and **vii)** ADP (Pang et al., 2019) a model diversification approach tested with white-box attacks in **Appendix** J. For baselines, we use undefended single and ensemble models that make predictions using all models.

**Evaluation Metrics.** Notably, with a defense employing randomness, the same input can result in different outputs (e.g. different scores) and even correct or incorrect predictions. Thus, when an adversarial input created by an attack aims to fool a model, it could fail or succeed. The more frequently it fails, the more robust the randomness defence is. Hence, we define the robustness of a randomness-based defense as follows:

$$\text{Robustness} = \mathbb{E}_{\boldsymbol{x}_{\text{adv}} \sim \boldsymbol{\mathcal{D}}_{\text{ADV}}}[\text{Acc}_r(\boldsymbol{x}_{\text{adv}})], \tag{12}$$

where $\text{Acc}_r(x_{\text{adv}})$ is the number of correct predictions over 1000 predictions of an adversarial example. $\mathcal{D}_{\text{ADV}}$ is a set of adversarial examples generated by an attack.

**Evaluation Protocol.** Recall, when a benign input is fed to a model incorporating randomness, it can be correctly or incorrectly classified. The more frequently a benign input is misclassified, the less reliable the input will be for the purpose of constructing an attack. Although it significantly increases the computation burden, for a fair and reliable comparison, we select benign inputs correctly inferred over *1,000* repeated queries, dubbed *reliable benign inputs*. To manage the computational burden on three different tasks, we compose each evaluation set with 500 reliable benign inputs and use a budget of 10K queries for each attack. For our method, we train a set of $K = 40$ models for `MNIST` task and $K = 10$ models for `CIFAR-10` and `STL-10` tasks and randomly select a subset of $N = 5$ models to make predictions. Other selection schemes and results are presented in **Appendix** K.

### 4.1 ROBUSTNESS AGAINST QUERY-BASED BLACK-BOX ATTACKS

We report robustness under **7** state-of-the-art attacks, consider all *three* perturbation objectives ($l_2$, $l_\infty$ $l_0$) and include decision and score-based attacks. We evaluate ours and **5** defenses, including adversarial training, with some performance evaluations deferred to the **Appendices M–J**.

Table 1: $l_2$ **objective.** Robustness (higher $\uparrow$ is stronger) of different defense methods against SIGNHUNTER and SQUAREATTACK.

| | CIFAR-10 | | | | | | | | | |
|---|---|---|---|---|---|---|---|---|---|---|
| | $l_2 =0.8$ | 1.6 | 2.4 | 3.2 | 4.0 | $l_2 =0.8$ | 1.6 | 2.4 | 3.2 | 4.0 |
| Single (*undef*) | 0.2% | 0.0% | 0.0% | 0.0% | 0.0% | 0.2% | 0.0% | 0.0% | 0.0% | 0.0% |
| Ensemble (*undef*) | 15.6% | 1.2% | 0.2% | 0.2% | 0.2% | 7.8% | 0.0 % | 0.0% | 0.0% | 0.0% |
| RND | 99.93% | 98.7 % | 93.75% | 84.1% | 73.81% | 99.03% | 87.49 % | 68.68% | 49.41% | 34.73% |
| RF | **99.98%** | 99.2 % | 94.24% | 85.2% | 75.14% | 99.5% | 91.14 % | 72.34% | 52.31% | 39.59% |
| **Disco** | 99.96% | **99.25%** | **97.61%** | **93.63%** | **90.24%** | **99.56%** | **95.62%** | **87.07%** | **76.5%** | **65.76%** |
| | STL-10 | | | | | | | | | |
| | $l_2 =1.2$ | 2.4 | 3.6 | 4.8 | 6.0 | $l_2 =1.2$ | 2.4 | 3.6 | 4.8 | 6.0 |
| Single (*undef*) | 27.0% | 2.8% | 0.6% | 0.0% | 0.0% | 24.6% | 1.8% | 0.6% | 0.0% | 0.0% |
| Ensemble (*undef*) | 46.2% | 11.4% | 3.0% | 1.0% | 0.6% | 43.0% | 6.6% | 1.0% | 0.4% | 0.2% |
| RND | 99.98% | 99.68 % | 98.92% | 97.19% | 92.74% | 99.93% | 98.63 % | 94.32% | 87.8% | 80.78% |
| RF | 99.99% | 99.63 % | 99.21% | 97.2% | 93.86% | 99.88% | 99.44 % | 97.53% | 95.06% | 89.67% |
| **Disco** | **99.99%** | **99.96%** | **99.8%** | **99.39%** | **98.75%** | **99.97%** | **99.74%** | **98.76%** | **96.86%** | **93.86%** |

Table 2: $l_0$ **objective**. Robustness (higher ↑ is stronger) of defenses against SPARSERS with `CIFAR-10` task.

| Methods | $l_0$ =16px | 32px | 48px | 64px |
|---|---|---|---|---|
| Single (*undef*) | 0.0% | 0.0% | 0.0% | 0.0% |
| Ensemble (*undef*) | 1.6% | 0.0% | 0.0% | 0.0% |
| RND | 45.27% | 23.77% | 15.88% | 11.98% |
| RF | 38.68% | 24.35 % | 20.66% | 15.89% |
| **Disco** | **63.85%** | **47.84%** | **41.59%** | **36.81%** |

Table 3: *Decision-based* **attacks.** Robustness of different defense methods against HOPSKIPJUMP ($l_2$) and SPAEVO ($l_0$) with the `CIFAR-10` task.

| Methods | HOPSKIPJUMP | | | | SPAEVO | | | |
|---|---|---|---|---|---|---|---|---|
| | $l_2$ =0.8 | 1.6 | 2.4 | 3.2 | $l_0$ =4px | 8px | 12px | 16px |
| Single (*undef*) | 0.0% | 0.0% | 0.0% | 0.0% % | 43.2% | 19.6% | 7.2% | 3.8% |
| Ensemble (*undef*) | 3.2% | 0.2% | 0.2% | 0.2% | 59.2% | 29.6% | 17.2% | 8.4% |
| RND | 99.94% | 99.13 % | 98.23% | 96.71% | 93.45% | 93.08% | 92.86% | 92.83% |
| RF | 99.91% | 98.53 % | 97.13% | 95.57% | 91.84% | 91.49% | 91.4% | 91.4% |
| **Disco** | **99.94%** | **99.4%** | **98.77%** | **98.04%** | **96.17%** | **95.99%** | **95.94%** | **95.88%%** |

*Performance Against Score-Based Attacks.* We report the performance of defense methods against score-based attacks SIGNHUNTER ($l_2$) and SQUAREATTACK ($l_2$) on three different tasks in Table 1. For `CIFAR-10` and `STL-10`, we configure five out of 10 models. The results in SIGNHUNTER and SQUAREATTACK are strong adversarial attacks. The results demonstrate our method consistently outperforms *state-of-the-art* defenses across tasks and perturbation budgets. This empirical evidence supports our theoretical analysis in Section 3. We provide further evidence, with additional results using different configurations for model randomization and for `MNIST` in **Appendix K**.

Table 4: $l_\infty$ **objective**. Robustness (higher ↑ is stronger) of different defense methods against NE-SATTACK, SIGNHUNTER and SQUAREATTACK with the `CIFAR-10` task.

| Attack | Methods | $l_\infty$ =0.02 | 0.04 | 0.06 | 0.08 | 0.1 |
|---|---|---|---|---|---|---|
| NESATTACK | Single (*undef*) | 82.8% | 62.0% | 41.2% | 26.8% | 15.4% |
| | Ensemble (*undef*) | 91.2% | 76.6% | 58.6% | 45.0% | 31.6% |
| | RND | 99.69% | 96.03 % | 90.94% | 84.75% | 77.83% |
| | RF | 99.5% | 95.57 % | 88.69% | 85.55% | 79.86% |
| | **Disco** | **99.7%** | **97.93%** | **94.39%** | **90.5%** | **86.77%** |
| SIGNHUNTER | Single (*undef*) | 1.8% | 0.0% | 0.0% | 0.0% | 0.0% |
| | Ensemble (*undef*) | 29.46% | 0.6% | 0.0% | 0.0% | 0.0% |
| | RND | 99.98% | 98.27 % | 88.97% | 75.63% | 63.73% |
| | RF | **99.99%** | 98.51 % | 87.38% | 72.23% | 61.5% |
| | **Disco** | 99.97% | **98.95%** | **95.56%** | **90.7%** | **84.22%** |
| SQUAREATTACK | Single (*undef*) | 2.2% | 0.0% | 0.0% | 0.0% | 0.0% |
| | Ensemble (*undef*) | 28.8% | 1.2% | 0.2% | 0.2% | 0.2% |
| | RND | 99.96% | 90.43 % | 63.68% | 39.44% | 22.06% |
| | RF | 99.92% | 88.97 % | 63.4% | 40.25% | 25.03% |
| | **Disco** | **99.97%** | **96.91%** | **86.52%** | **70.22%** | **55.77%** |

We further evaluate the robustness of defenses against 3 strong, $l_\infty$ attacks, NESATTACK, SIGN-HUNTER and SQUAREATTACK as well as $l_0$ attack SPARSERS. The results in Tables 2 and 4 show that our model randomization mechanism is more robust than random noise injection defenses across different attacks and perturbation objectives.

*Performance Against Decision-Based Attacks.* We report results for HOPSKIPJUMP ($l_2$) and SPAEVO ($l_0$) attacks in Table 3. Our proposed defense demonstrates stronger robustness than the *state-of-the-art* defenses across different decision-based attacks and perturbation objectives. Importantly, the empirical evidence supports our theoretical analysis in Section 3.

## 4.2 ROBUSTNESS AGAINST ADAPTIVE ATTACKS

We compare RND, RF with our method under *adaptive* SIGNHUNTER and *adaptive* SQUAREATTACK employing Expectation Over Transformation (EOT) (Athalye et al., 2017). Our explanation of EOT-based adaptive attacks is presented in Appendix A. Figure 2 shows that an *adaptive* attacker can alleviate the effect of defense mechanisms compare to their *non-adaptive* counterparts but with a $10\times$ higher cost for queries. Interestingly, our insights into obfuscating the relationship between query-response pairs with model randomization outperform random noise injection methods.

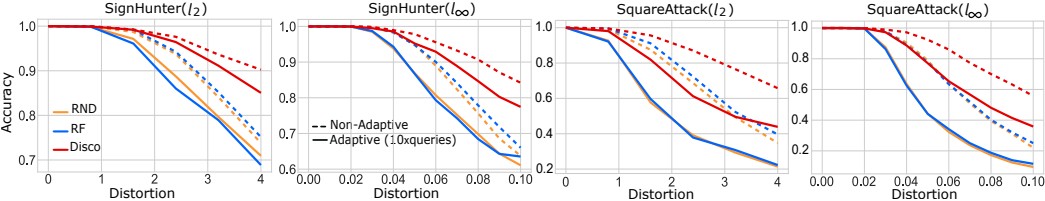

Figure 2: Robustness against **adaptive** vs. **non-adaptive** $l_2$ and $l_\infty$ objective attacks using SIGN-HUNTER and SQUAREATTACK. For *adaptive* attacks, the adversary expends extra, $m = 10\times$ queries for each input, and averages the outputs to mitigate obfuscation (we defer details to **Appendix A**).

Further, we evaluate ours against two additional adaptive attacks in **Appendix A.2** and **A.3**.

## 4.3 INVESTIGATING CLEAN ACCURACY OF UNDEFENDED AND DEFENDED MODELS

Defenses invariably compromise clean accuracy for robustness. We report clean accuracy achieved by undefended and defended models along with the resulting clean accuracy drop (CAD) denoted by ($\downarrow \Delta$) across the tasks in Tables 5. Importantly, our goal to seek *well-performing* models with the

incorporation of the new learning objective in Section 3.6.2 has mitigated the CAD drop significantly better than state-of-the-art random noise defenses. We provide further evidence to demonstrate the impact of the learning objective in **Appendix** D.

Table 5: **Clean accuracy achieved by undefended and defended models**. For our method, SVGD+ (All) is the clean accuracy of the entire model set while Disco(SVGD+) presents the clean accuracy of random five out of 40 models (MNIST) or random five out of 10 models (CIFAR-10, STL-10)—we report clean accuracy for other model randomization configurations in **K.1**.

| Dataset | Baselines | | Defense Methods | | | Ours | |
|---|---|---|---|---|---|---|---|
| | Single Model | Ensembles | RND ($\downarrow \Delta$) | RF ($\downarrow \Delta$) | | SVGD+ (All) | Disco(SVGD+) ($\downarrow \Delta$) |
| MNIST | 99.64% | 99.72% | 98.59% ($\downarrow$1.05%) | 98.45% ($\downarrow$1.19%) | | 99.59% | 99.34% ($\downarrow$**0.3%**) |
| CIFAR-10 | 92.09% | 94.76% | 87.63% ($\downarrow$4.46%) | 89.73% ($\downarrow$2.36%) | | 93.19% | 92.26% ($\uparrow$**0.87%**) |
| STL-10 | 90.39% | 92.15% | 86.38% ($\downarrow$4.01%) | 88.5% ($\downarrow$1.89%) | | 90.18% | 88.97% ($\downarrow$**1.4%**) |

## 4.4 ROBUSTNESS OF DIVERSITY PROMOTING ALTERNATIVES

Table 6: Robustness (higher $\uparrow$ is stronger) of diversity promotion approaches with Disco against SQUAREATTACK ($l_2$ objective) with the CIFAR-10 task (further results are in **Appendix** I).

| Methods | $l_2 =0.8$ | 1.6 | 2.4 | 3.2 | 4.0 |
|---|---|---|---|---|---|
| Disco(Ensemble) | 98.2% | 87.9 % | 76.9% | 63.5% | 52.2% |
| Disco(DivDis) | 99.1% | 94.1 % | 82.7% | 70.4% | 56.4% |
| Disco(DivReg) | 99.2% | 90.9 % | 76.3% | 64.6% | 51.6% |
| **Disco(SVGD+)** | **99.56%** | **95.62%** | **87.07%** | **76.5%** | **65.76%** |

We assess alternative methods for promoting model diversity (Deep Ensembles, Ensemble, (Lakshminarayanan et al., 2017); DivDis (Lee et al., 2023); DivReg (Teney et al., 2022)) to: i) evaluate their performance under our model randomization method; and ii) understand the relationship between robustness and model diversity. We defer formulations of these training objectives to **Appendix** H.1.

*Performance.* We compare robustness against score-based, $l_2$ adversarial attacks SIGNHUNTER and SQUAREATTACK. The results in Table 6.

> The results show that our diversity objectives outperforms the alternatives. Generally, explicit diversity promotion objectives achieves improved robustness compared to Ensemble (see additional results on diversity analysis and additional attacks in **Appendix** I.

## 4.5 ROBUSTNESS, COST AND IMPLEMENTATION IN LARGE-SCALE TASKS (OPENCLIP)

Unsurprisingly, higher robustness and model utility leads to an increase in overhead (training time, inference time and model storage). We analyze overheads in Appendix E. In general, any overhead is undesirable when provisioning models. Therefore, we adopt low-rank learning (LoRA) to significantly mitigate the overhead and provide empirical results with the high-resolution ImageNet task with a set of five large-scale OPENCLIP models using a two out of five randomization for responses.

*Performance.* Now the overhead of building an additional model in trainable parameters is $< 0.4\%$, even *without* parallelization, inference time scales only linearly as seen in Table 7. Results in Table 8 demonstrate Disco to remain the most *effective* defense. We defer our detailed analysis to **Appendix** F.

Table 7: Parameters, Storage and Inference for five OpenCLIP with LoRA trained on ImageNet.

| Models | Single CLIP | CLIP (LoRA) Model Set |
|---|---|---|
| Trainable Parameters | 114 M | 1.84 M ($\uparrow$1.6%, 0.32% per model) |
| Storage | 433 MB | 439 MB ($\uparrow$1.38%, 0.28% per model) |
| Inference time | 4.4 ms | 9 ms (sequential, 2 of 5) |

Table 8: Robustness of defenses against SQUAREATTACK ($l_\infty$ objective) on the ImageNet task with OpenCLIP models.

| Methods | $l_\infty =0.025$ | 0.05 | 0.075 | 0.1 |
|---|---|---|---|---|
| RND | 83.39% | 61.95% | 43.37% | 24.89% |
| RF | 86.45% | 65.1 % | 51.14% | 35.83% |
| **Disco** | **90.76%** | **72.51%** | **56.17%** | **45.4%** |

## 5 CONCLUSION

This study investigates the effectiveness of a model randomization defense against query-based black-box attacks in both score-based and decision-based settings. We theoretically analyze the defense and prove the link between diversity of model outputs and model robustness. We realize the approach by learning a set of diverse yet well-performing models for random selection to provide robustness whilst minimizing the clean accuracy drop of defended models. We demonstrate the approach leads to an effective, practical defense with 7 state-of-the-art query-based black-box attacks under all *three* perturbation objectives ($l_\infty, l_2, l_0$).

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

## CONTENTS IN THE APPENDIX

We provide a brief overview of the additional experimental results and findings in the Appendices that follow.

1. Proofs for theoretical analysis against gradient estimation attacks (Appendix A) and empirical evaluation against two additional adaptive attacks (Appendix A.2 and A.3) and gradient-free (Appendix B).

2. Analysis of Trade-off Between Subset Set Size and Error Estimation C.

3. Effectiveness of the proposed learning objective in Section 3.6.2 (Appendix D).

4. Analysis of the overhead from our model randomization method (Appendix E)

5. Ameliorating the overhead for large-scale tasks and evaluations with the large-scale Open-CLIP model and the `ImageNet` task (Appendix F).

6. Effectiveness against a stronger attack in a *surrogate* model setting (Appendix G).

7. Formulations of and diversity analysis of alternative approaches for promoting model diversity (Appendix H).

8. Robustness evaluations using alternative approaches for model diversity promotion with Disco against **4** state-of-the-art attacks under $l_\infty$ and $l_0$ perturbation objectives (Appendix I).

9. Robustness comparison with Adaptive Diversity Promoting (ADP) method (Appendix J).

10. Robustness and clean accuracy evaluations of different randomized model selection strategies with different model diversification methods (Appendix K).

11. Robustness over multiple trials (Monte Carlo experiment) (Appendix L).

12. Robustness comparisons with **4** additional defenses:
    - Adversarial Training (AT) defense (Appendix M).
    - Adversarial Attack on Attackers (AAA) Defense (Appendix N).
    - Adding Noise to The Output Scores (Appendix O
    - Region-Based Classification (RBC) (Appendix P).

13. Fundamental differences from PuriDefense (Appendix Q)

14. Discussion, limitation and broader impacts of our research (Appendix R).

# A  PROOF FOR THEORETICAL ANALYSIS AGAINST GRADIENT ESTIMATION ATTACKS

In this section, we provide the theoretical analysis of our defense method against gradient-estimation attacks and the proof of proposition 3.1.

*Proof.* We consider gradient estimation when the entire set of models (or even a single model) is presented to the attacker versus the expectation of gradient estimation under different subsets.

Given an input $\boldsymbol{x} \in \mathbb{R}^d$ and $K$ models, the output logits of $K$ models is $F(\boldsymbol{x}) = \frac{1}{K}\sum_{k=1}^{K} f(\boldsymbol{x};\boldsymbol{\theta}_k)$, where $\mathcal{F} = \{f(\cdot, \boldsymbol{\theta}_1), f(\cdot, \boldsymbol{\theta}_2), \ldots, f(\cdot, \boldsymbol{\theta}_K)\}$. Given $\epsilon > 0$, $\boldsymbol{u} \sim \mathcal{N}(0, \boldsymbol{I})$, $\boldsymbol{u} \in \mathbb{R}^d$, the *gradient estimated* when the entire model set is used can be formulated as follows:

$$\hat{G}(\boldsymbol{x}) = \mathbb{E}_{\boldsymbol{u}}\left[\frac{F(\boldsymbol{x} + \epsilon\boldsymbol{u}) - F(\boldsymbol{x})}{\epsilon}\boldsymbol{u}\right]; \ \ \hat{G}(\boldsymbol{x}) \in \mathbb{R}^d.$$

Applying Taylor expansion at $\boldsymbol{x}$, we have $F(\boldsymbol{x} + \epsilon\boldsymbol{u}) \approx F(\boldsymbol{x}) + \epsilon\boldsymbol{u}\nabla F(\boldsymbol{x})$. Then, we have:

$$\hat{G}(\boldsymbol{x}) \approx \mathbb{E}_{\boldsymbol{u}}\left[\frac{F(\boldsymbol{x}) + \epsilon\boldsymbol{u}\nabla F(\boldsymbol{x}) - F(\boldsymbol{x})}{\epsilon}\boldsymbol{u}\right] \approx \mathbb{E}_{\boldsymbol{u}}[\boldsymbol{u}\nabla F(\boldsymbol{x})\boldsymbol{u}]$$

$$\approx \mathbb{E}_{\boldsymbol{u}}\left[\boldsymbol{u}\frac{1}{K}\left[\sum_{k=1}^{K}\nabla f(\boldsymbol{x}, \boldsymbol{\theta}_k)\right]\boldsymbol{u}\right]$$

Notably, since $\boldsymbol{u}$ is sampled from a normal distribution, it provides an unbiased estimation of the gradient.

However, under our defense, a subset of models is sampled uniformly at random with replacement from $\mathcal{F}$ and an average is taken to make a prediction. Particularly, we sample $q(\boldsymbol{x}; \boldsymbol{\pi}) = \frac{1}{N}\sum_{k=1}^{N}\pi_k f(\boldsymbol{x}, \boldsymbol{\theta}_k)$ where $\boldsymbol{\pi} \sim \mathcal{B}(\mu_1, \ldots, \mu_K)$ denotes a $K$ dimensional vector sampled from $K$ independent Bernoulli distributions and $N$ is the size of the model subset. Thus, the expectation of the estimated gradient from all subsets of models can be formulated as the following:

$$\tilde{G}(\boldsymbol{x}) = \mathbb{E}_{\boldsymbol{u}}\left[\mathbb{E}_{\pi}\left[\frac{q(\boldsymbol{x} + \epsilon\boldsymbol{u}; \boldsymbol{\pi}^{(i)}) - q(\boldsymbol{x}; \boldsymbol{\pi}^{(j)})}{\epsilon}\right]\boldsymbol{u}\right], \ \ \tilde{G}(\boldsymbol{x}) \in \mathbb{R}^d.$$

where $i, j$ denotes $i$- and $j-$th consecutive iterations (model queries). Applying Taylor expansion at $\boldsymbol{x}$, we have $q(\boldsymbol{x} + \epsilon\boldsymbol{u}; \boldsymbol{\pi}^{(i)}) \approx q(\boldsymbol{x}; \boldsymbol{\pi}^{(i)}) + \epsilon\boldsymbol{u}\nabla q(\boldsymbol{x}; \boldsymbol{\pi}^{(i)})$. Then, we have:

$$\tilde{G}(\boldsymbol{x}) \approx \mathbb{E}_{\boldsymbol{u}}\left[\mathbb{E}_{\pi}\left[\frac{q(\boldsymbol{x}; \boldsymbol{\pi}^{(i)}) + \epsilon\boldsymbol{u}\nabla q(\boldsymbol{x}; \boldsymbol{\pi}^{(i)})] - q(\boldsymbol{x}; \boldsymbol{\pi}^{(j)})}{\epsilon}\right]\boldsymbol{u}\right]$$

$$\approx \mathbb{E}_{\boldsymbol{u}}\left[\mathbb{E}_{\pi}\left[\frac{q(\boldsymbol{x}; \boldsymbol{\pi}^{(i)}) - q(\boldsymbol{x}; \boldsymbol{\pi}^{(j)}) + \epsilon\boldsymbol{u}\nabla q(\boldsymbol{x}; \boldsymbol{\pi}^{(i)})}{\epsilon}\right]\boldsymbol{u}\right]$$

$$\approx \mathbb{E}_{\boldsymbol{u}}\left[\mathbb{E}_{\pi}\left[\frac{q(\boldsymbol{x}; \boldsymbol{\pi}^{(i)}) - q(\boldsymbol{x}; \boldsymbol{\pi}^{(j)})}{\epsilon} + \boldsymbol{u}\nabla q(\boldsymbol{x}; \boldsymbol{\pi}^{(i)})\right]\boldsymbol{u}\right]$$

$$\approx \mathbb{E}_{\boldsymbol{u}}\left[\left[\frac{1}{\epsilon}\left[\underbrace{\mathbb{E}_{\pi}\left[q(\boldsymbol{x}; \boldsymbol{\pi}^{(i)})\right]}_{A} - \underbrace{\mathbb{E}_{\pi}\left[q(\boldsymbol{x}; \boldsymbol{\pi}^{(j)})\right]}_{B}\right] + \boldsymbol{u}\mathbb{E}_{\pi}\left[\nabla q(\boldsymbol{x}; \boldsymbol{\pi}^{(i)})\right]\right]\boldsymbol{u}\right]$$

Since:

$$\mathbb{E}_{\pi}\left[q(\boldsymbol{x}; \boldsymbol{\pi}^{(i)})\right] = \sum_{k}^{K}\mu_k f(\boldsymbol{x}; \boldsymbol{\theta}_k),$$

and the difference between the first two expectation terms A and B will approach zero, we have:

$$\tilde{G}(\boldsymbol{x}) \approx \mathbb{E}_{\boldsymbol{u}}\left[\boldsymbol{u}\mathbb{E}_{\pi}\left[\nabla q(\boldsymbol{x}; \boldsymbol{\pi}^{(i)})\right]\boldsymbol{u}\right] \approx \mathbb{E}_{\boldsymbol{u}}\left[\boldsymbol{u}\left[\sum_{k}^{K}\mu_k\nabla f(\boldsymbol{x}; \boldsymbol{\theta}_k)\right]\boldsymbol{u}\right]$$

Thus, we have:

$$\tilde{G}(\boldsymbol{x}) - \hat{G}(\boldsymbol{x}) \approx \mathbb{E}_{\boldsymbol{u}}\left[\boldsymbol{u}\left[\sum_{k=1}^{K}(\mu_k - \frac{1}{K})\nabla f(\boldsymbol{x};\boldsymbol{\theta}_k)\right]\boldsymbol{u}\right].$$

As $\mu_k$ approaches $\frac{1}{K}$, the difference between $\tilde{G}(\boldsymbol{x})$ and $\hat{G}(\boldsymbol{x})$ approaches zero.

*Given the result above, first we consider non-adaptive attackers.*

**Non-adaptive attack setting.** Generally, an adversary does not have knowledge of defense mechanisms. Hence, under our defense mechanism, the gradient estimator with a pair of samples is:

$$g(\boldsymbol{x}) = \frac{q(\boldsymbol{x} + \epsilon\boldsymbol{u};\boldsymbol{\pi}^{(2)}) - q(\boldsymbol{x};\boldsymbol{\pi}^{(1)})}{\epsilon}\boldsymbol{u}, \;\; g(\boldsymbol{x}) \in \mathbb{R}^d.$$

In practice, to achieve a good approximation of gradient, the finite difference method samples multiple $\boldsymbol{u}$, obtains multiple $g(\boldsymbol{x})$ and takes the average. Then, the approximation of the gradient with $n$ different pairs of samples using the finite difference method is formulated as follows:

$$\bar{g}(\boldsymbol{x}) = \frac{1}{n}\sum_{i=1}^{n}\frac{q(\boldsymbol{x} + \epsilon\boldsymbol{u}_i;\boldsymbol{\pi}^{(2i)}) - q(\boldsymbol{x};\boldsymbol{\pi}^{(2i-1)})}{\epsilon}\boldsymbol{u}_i, \;\; \bar{g}(\boldsymbol{x}) \in \mathbb{R}^d.$$

where the defender generates $\boldsymbol{\pi}^{(2i)}, \boldsymbol{\pi}^{(2i-1)} \sim \mathcal{B}(\mu_1, \ldots, \mu_K)$, the attacker generates $\boldsymbol{u}_i \sim \mathcal{N}(0, \boldsymbol{I})$. However, there is a gap between this approximation $\bar{g}(\boldsymbol{x})$ and the *expected gradient estimation* of all subsets $\tilde{G}(\boldsymbol{x}) = \mathbb{E}[g(\boldsymbol{x})]$. Since we proved the *expected gradient estimation* $\tilde{G}(\boldsymbol{x})$ approximates the *actual gradient estimation* of the entire model set $\hat{G}(\boldsymbol{x})$, $|\bar{g}(\boldsymbol{x}) - \tilde{G}(\boldsymbol{x})|$ approximates to $|\bar{g}(\boldsymbol{x}) - \hat{G}(\boldsymbol{x})|$. If each element of the gradient $g(\boldsymbol{x})$ estimated at iteration $i$ is bounded by $a_i^j \leq g(\boldsymbol{x})^j \leq b_i^j$ with $\boldsymbol{a}_i, \boldsymbol{b}_i \in \mathbb{R}^d$, $n$ different gradient estimators $g(.)$ are independent random variables and $A^j$ is defined as $|\bar{g}(x)^j - \hat{G}(x)^j| \geq \Delta$, according to the Hoeffding's inequality and employing a union bound over all $d$ dimensions to bound the probability of deviation in any component, we have:

$$P(\cup_{j=1}^{d}A^j) \leq \sum_{j=1}^{d}P(A^j) = \sum_{j=1}^{d}2\exp\left(-\frac{2n^2\Delta^2}{\sum_{i=1}^{n}(a_i^j - b_i^j)^2}\right).$$

Where $\Delta$ is a gap or margin of error. This term can further be upper bounded by considering the fact that $\exp(-x)$ is monotonically decreasing, we know for any $j$:

$$\exp\left(-\frac{2n^2\Delta^2}{\sum_{i=1}^{n}(a_i^j - b_i^j)^2}\right) \leq \exp\left(-\frac{2n^2\Delta^2}{\sum_{i=1}^{n}[\max_j(a_i^j - b_i^j)^2]}\right)$$

Therefore, we have:

$$P(\cup_{j=1}^{d}A^j) \leq \sum_{j=1}^{d}2\exp\left(-\frac{2n^2\Delta^2}{\sum_{i=1}^{n}(a_i^j - b_i^j)^2}\right) \leq 2d\exp\left(-\frac{2n^2\Delta^2}{\sum_{i=1}^{n}[\max_j(b_i^j - a_i^j)]^2}\right)$$

To achieve low margin of error $\Delta$, the upper bound of the probability such that this gap is beyond $\Delta$ must be low. To achieve this with the desired confidence level $1 - \delta$ and the given bound as above, we set the right-hand side of the inequality smaller than $\delta$ and solve for $n$ as the following:

$$2d\exp\left(-\frac{2n^2\Delta^2}{\sum_{i=1}^{n}[\max_j(b_i^j - a_i^j)]^2}\right) \leq \delta$$

$$-\frac{2n^2\Delta^2}{\sum_{i=1}^{n}[\max_j(b_i^j - a_i^j)]^2} \leq \log\frac{\delta}{2d}$$

$$\frac{2n^2\Delta^2}{\sum_{i=1}^{n}[\max_j(b_i^j - a_i^j)]^2} \geq \log\frac{2d}{\delta}$$

$$n^2 \geq \frac{\log\frac{2d}{\delta}\sum_{i=1}^{n}[\max_j(b_i^j - a_i^j)]^2}{2\Delta^2}$$

$$n \geq \sqrt{\frac{\log\frac{2d}{\delta}\sum_{i=1}^{n}[\max_j(b_i^j - a_i^j)]^2}{2\Delta^2}}$$

This implies that when a set of models is more diverse, the bound $a_i^j < g(\boldsymbol{x})^j < b_i^j$ is larger, and the number of samples $n$ needed, such that every element of $\bar{g}(\boldsymbol{x}) - \hat{G}(\boldsymbol{x})$ is more likely within the error margin $\Delta$, grows significantly. ∎

*Next we consider adaptive attackers.*

**Adaptive attack setting.** Now, we assume the attacker has knowledge of the defense mechanism and is aware that a subset of $K$ models is randomly selected to generate the response to a model query. If an adversary has prior knowledge of our defense mechanism, they can employ Expectation Over Transformation (EOT) Athalye et al. (2017) to obtain a more accurate gradient estimate.

It is worth noting that, in the original EOT method, the adversarial perturbation gradient is calculated based on a series of transformed inputs (adversarial examples). The reason is to address the issue incurred by the ineffectiveness of an adversarial example yielded by an adversary when the adversarial example is randomly transformed (Athalye et al., 2017) *i.e.* with different view angles. Therefore, to maintain the effectiveness of an adversarial example over different transformations, they model these transformations in their optimization procedure by transforming the inputs. Similarly, to maintain the effectiveness of an adversarial example over different models whose selection is represented by different $\boldsymbol{\pi}$, an adversary seeks a perturbation gradient direction such that it is effective over different models. Therefore, in our study, the so-called EOT is performed over $\boldsymbol{\pi}$. Interestingly, the same reasoning is in Athalye et al. (2018) and Nguyen et al. (2024) when applying EOT to attack defenses involving stochasticity, like with ours.

In practice, similar to a non-adaptive attack, an EOT-based adaptive attack sends $m$ queries to a target model to estimate the gradient, but for each query, it feeds a target model with the same input $n$ times to mitigate the impact of randomness. As a result, the number of queries to estimate a gradient in the adaptive setting is $m \times n$. This is $m\times$ higher than a non-adaptive attack. For instance, if a non-adaptive attack uses 10K queries and $m = 10$, the total number of queries needed by an adaptive attacker is 100K. Likewise, for gradient-free attacks, each input is fed into a target model $m$ times to find a more reliable attack direction.

Formally, under our defense mechanism, the gradient estimator employed by an adaptive attack using the finite difference method is formulated as follows:

$$g'(\boldsymbol{x}) = \frac{1}{m} \sum_1^m \frac{f^{(i)}(\boldsymbol{x} + \epsilon \boldsymbol{u}; \boldsymbol{\pi}^{(i)}) - f^{(j)}(\boldsymbol{x}; \boldsymbol{\pi}^{(j)})}{\epsilon} \boldsymbol{u} = \frac{1}{m} \sum_1^m g(\boldsymbol{x})$$

where $\boldsymbol{\pi}^{(i)}, \boldsymbol{\pi}^{(j)} \sim \mathcal{B}(\mu_1, \ldots, \mu_K)$ is generated by the defender, $\boldsymbol{u} \sim \mathcal{N}(0, \boldsymbol{I})$ is generated by the attacker and $f^{(i)}(\boldsymbol{x}; \boldsymbol{\pi}^{(i)}) = \sum_k \boldsymbol{\pi}_k^{(i)} f_k(\boldsymbol{x}, \boldsymbol{\theta}_k)$, $f^{(j)}(\boldsymbol{x}; \boldsymbol{\pi}^{(j)}) = \sum_k \boldsymbol{\pi}_k^{(j)} f_k(\boldsymbol{x}, \boldsymbol{\theta}_k)$.

Similar to the non-adaptive setting, to achieve a good approximation of gradient, the finite difference method samples multiple $\boldsymbol{u}$, obtains multiple $g'(\boldsymbol{x})$ and takes the average $\tilde{g}(\boldsymbol{x}) = \frac{1}{n} \sum_1^n g'(\boldsymbol{x})$. Then, we have $\tilde{g}(\boldsymbol{x}) = \frac{1}{n'} \sum_1^n \sum_1^m g(\boldsymbol{x})$ with $n' = n \times m$. If each element of the gradient $g'(\boldsymbol{x})$ estimated at iteration $i$ is bounded by $a_i'^j \leq g'(\boldsymbol{x})^j \leq b_i'^j$ with $\boldsymbol{a}_i', \boldsymbol{b}_i' \in \mathbb{R}^d$, and $A'^j$ is defined as $|\tilde{g}(x)^j - \hat{G}(x)^j| \geq \Delta$, according to the Hoeffding's inequality and employing a union bound over all $d$ dimensions to bound the probability of deviation in any component, we have:

$$P(\cup_{j=1}^d A'^j) \leq \sum_{j=1}^d P(A'^j) = \sum_{j=1}^d 2 \exp\left(-\frac{2n'^2 \Delta^2}{\sum_{i=1}^n (a_i'^j - b_i'^j)^2}\right)$$

The number of samples $n'$ needed to ensure every element of $\tilde{g}(\boldsymbol{x}) - \hat{G}(\boldsymbol{x})$ more likely within an error margin $\Delta$ with confidence $1 - \delta$ is at least:

$$n' \geq \sqrt{\frac{\log \frac{2d}{\delta} \sum_{i=1}^n [\max_j (b_i'^j - a_i'^j)]^2}{2\Delta^2}}$$

This implies that the number of samples $n'$ relies on the range of estimator $d'$ with a given confidence interval and margin of error. Importantly, as similar to non-adaptive attacks, when a set of models is more diverse, the bound $a_i'^j \leq g'(\boldsymbol{x})^j \leq b_i'^j$ is larger, and the number of samples $n'$ needed, such that every element of $\tilde{g}(\boldsymbol{x}) - \hat{G}(\boldsymbol{x})$ is more likely within the error margin $\Delta$, grows significantly.

Interestingly, in adaptive settings, when sampling each $\boldsymbol{u}$, the attacker has to sample $m$ times with the same $\boldsymbol{u}$. Thus, the total number of samples an adaptive attacker needs is significantly higher; since $n' = n \times m$.

## A.1 THE INSIGHT FROM PROPOSITION 3.1

The primary purpose of the proposition 3.1 is to formalize the relationship between model diversity and the query complexity of the attack. Because our investigation revolves around model randomisation where models are sampled from a set of diverse models to defend against black-box attacks.

- The key insight lies in the range bounds $a_i$ and $b_i$ of gradient estimators. These bounds capture how much the average gradient estimator $\bar{g}$ can deviate from the estimated gradient of the entire set of models $\hat{G}$ and thus how "spread out" or diverse the model outputs are. In this way, greater diversity across models—regardless of how many models there are—leads to larger $b_i - a_i$ ranges, which in turn implies a higher number of queries $n$ required for the attack to accurately estimate the ensemble behavior. This reflects a key benefit of diversity and amplifies the attacker's sampling complexity.

- While $b_i$ and $a_i$ are not typically known in closed-form, they are not meant to be set or computed directly. Rather, they can be seen as analytic tools allowing us to formally capture and reason about how diversity across all individual models hinders attacks. The proposition is thus a theoretical lens for understanding the attack cost amplification induced by diversity in randomized ensembles. It also shows us that the cost of an attack can be made large when the underlying set of models is able to generate highly diverse outputs for given pairs of inputs.

- From the proposition, we can show how complex estimating the average gradient $\bar{g}$ correlates with the range of variation in the output scores rather than $K$ as quantified by $a_i$ and $b_i$. In a conventional setting, without our proposed defense mechanism, model parameters lack the deliberate diversification mechanism outlined in Section 3.6.1. This absence of systematic parameter divergence results in functionally similar models that exhibit limited diversity in their output score distributions. Thus, the gradient estimator bound range $b_i - a_i$ will tend to be small and a lower bound for the required number of queries $n$ in equation 7 is low to estimate a direction.

## A.2 THE WORST LOSS ADAPTIVE ATTACK

**Clarification about EOT Adaptive Attack.** While EOT style adaptive attacks do not cover all possible adaptive attacks, it is an effective means to mitigate the stochastic nature of a defense mechanism as demonstrated in Section 4.2, and used in prior work (Qin et al., 2021; Nguyen et al., 2024). A viable alternative is the worst-case selection adaptive attack, which we analyze and show the empirical results in the section below. Overall, our defense still withstands this new adaptive attack effectively.

**Analysis of a new adaptive attack.** A stronger adaptive attacker could attempt the worst-case loss strategy as described—querying the same input multiple times to probe the ensemble distribution, then selecting the model output that yields the worst-case loss for that input. However, this strategy will not works for our defense due to highly complex and non-smooth decision surface yielded by different combinations of diverse individual models. Concretely, the randomness in model sampling and model diversity leads to non-smooth decision surface. For instance, the decision surface of one subset of models is different from the other subsets and this differnce could be siginificant when diversity is high. Therefore, following the estimated gradient direction or search direction corresponding with the worst loss is possibly unreliable and inconsistent. For gradient estimation attacks, this can easily lead to entrapment in local minima as eluded in (Chen et al., 2020a; Vo et al., 2022b). For gradient-free attacks, they can also get trapped and this harmpers attack progress significantly.

It is, however, interesting to consider if this initial expense will actually translate into reaching an adversarial input. Notably, our approach selects a subset of models at random with replacement. The same set can be repeatedly sampled for the same input. Consequently, adversaries may not know

when the worst loss can be obtained even $N, K$ are known. The practical way is to select the worst case with a fixed number of times for sampling each point.

**Experiment results.** We designed a new adaptive attack which repeatedly sample each input (each point) 10 times (to compare with EOT style adaptive attack) and select the search direction which yield the worst loss. We use 10,000 iterations like 10,000 queries employed by non-adaptive attacks. This is equivelant to EOT style adaptive. Due to the time and resource constraints, the worst-loss strategy is adapted with SquareAttack ($l_2, l_\infty$) as we did with EOT style adaptive.

Our results in Table 9 show that the EOT style adaptive attack is stronger than worst-case selection adaptive attack. We highly suspect that the search direction yielded by EOT style adaptive is more reliable as it is less affected by non-smooth decision surface.

Table 9: Robustness (Lower is more robust) of different adaptive attacks ($l_2, l_\infty$) against Disco.

| Adaptive Methods | SQUAREATTACK ($l_2$) | | | | | SQUAREATTACK ($l_\infty$) | | | | |
|---|---|---|---|---|---|---|---|---|---|---|
| | $l_2$ =0.8 | 1.6 | 2.4 | 3.2 | 4.0 | $l_\infty$=0.2 | 0.4 | 0.6 | 0.8 | 1.0 |
| Worst Loss | 99.8 % | 98.21% | 92.45% | 86.2% | 78.99% | 99.97% | 98.11% | 86.49% | 70.12% | 55.06% |
| EOT style | **98.06**% | **81.88**% | **61.25**% | **49.57**% | **43.88**% | **99.77**% | **88.64**% | **65.32**% | **48.07**% | **35.86**% |

### A.3 A SPECIAL CASE OF THE WORST LOSS ADAPTIVE ATTACK WITH MATCHED SCORES FROM THE SAME MODEL AND $N = 1$)

**Experiment results.** Due to the complexity of finding the gradient of each possible subset of models (when $N > 1$), we will focus on the case when $N = 1$. We assume that the adversary could match the logit scores for $x$ and $x + \epsilon u$ from the same model and take the worst-case selection approach, corresponding to the worst loss reduction. In practice, we simplified this by allowing an attack to query and obtain scores for every pair of inputs ($x$ and $x + \epsilon u$) from the same individual model. To this end, we adopt SQUAREATTACK and follow the experiment settings employed in Appendix A.2 to achieve this goal. Particularly, we use models for CIFAR-10, use $K = 10$ and $N = 1$, with the adversary able to query and obtain gradient estimations for 10 selected models and select the worst. Our results in Table 10 show that under $N = 1$ setting with the worst case approach, the adaptive is stronger than EOT style adaptive attacks at low distortion budgets but it is less effective than EOT at high distortion levels.

**Analysis.** Highly complex and non-smooth decision surface. As discussed in Appendix A.2, the gradient selection strategy, even when $N = 1$ could be less effective as the decision surface is highly complex and different among individual models. This difference is high when trained models are diverse and results in a non-smooth decision surface. Therefore, following the estimated gradient direction or search direction corresponding to the worst loss is possibly unreliable and inconsistent. For instance, gradient or gradient-free attacks could get entrapped in local minima as eluded in (Chen et al., 2020a; Vo et al., 2022b).

Table 10: Robustness (Lower is more robust) of different adaptive attacks ($l_2, l_\infty$) against Disco.

| Adaptive Methods | SQUAREATTACK ($l_2$) | | | | | SQUAREATTACK ($l_\infty$) | | | | |
|---|---|---|---|---|---|---|---|---|---|---|
| | $l_2$ =0.8 | 1.6 | 2.4 | 3.2 | 4.0 | $l_\infty$=0.2 | 0.4 | 0.6 | 0.8 | 1.0 |
| Worst Loss | 99.8 % | 98.21% | 92.45% | 86.2% | 78.99% | 99.97% | 98.11% | 86.49% | 70.12% | 55.06% |
| EOT style | **98.06**% | **81.88**% | **61.25**% | **49.57**% | **43.88**% | **99.77**% | **88.64**% | **65.32**% | **48.07**% | **35.86**% |

Potentially overfit to the most vulnerable model. The worst-case selection adaptive attack could be overfit to the weakest or most vulnerable model in the set of models. Because the decision boundary of that single model may diverge significantly from those of other models, the resulting perturbation is unlikely to generalize across other models. In effect, the attacker might successfully fool one model but fail to mislead the rest, thereby undermining the effectiveness of the attack.

In general, our results and analysis are for when $N = 1$, but it could be generalised for subsets of models when $N > 1$. Therefore, as demonstrated in our results and those from other defence papers we use as a baseline, EOT style strategy remains a powerful adaptive attack and mostly yields higher attack success rates. Moreover, our results and analysis also underscore the robustness of our defence against the worst-case selection adaptive attacks in the setting $N = 1$.

## B   PROOF FOR THEORETICAL ANALYSIS AGAINST GRADIENT-FREE ATTACKS

*Proof.* Given input $\boldsymbol{x}$, a constant $\epsilon > 0$, $\boldsymbol{u} \sim \mathcal{N}0, \boldsymbol{I})$, the search direction of search-based attacks when considered against against the entire set of models (the ensemble or more generally a single model) relies on the sign of $\hat{H}(\boldsymbol{x}, \boldsymbol{u}) = F(\boldsymbol{x} + \epsilon\boldsymbol{u}) - F(\boldsymbol{x})$. Similarly, the search direction of search-based attacks against our defense employing different random subsets of models depends on the sign of $\tilde{H}(\boldsymbol{x}, \boldsymbol{u}) = q(\boldsymbol{x} + \epsilon\boldsymbol{u}; \boldsymbol{\pi}^{(i)}) - q(\boldsymbol{x}; \boldsymbol{\pi}^{(j)})$.

Applying Taylor expansion at $\boldsymbol{x}$, we have:

$$\hat{H}(\boldsymbol{x}, \boldsymbol{u}) = F(\boldsymbol{x} + \epsilon\boldsymbol{u}) - F(\boldsymbol{x}) \approx F(\boldsymbol{x}) + \epsilon\boldsymbol{u}\nabla F(\boldsymbol{x}) - F(\boldsymbol{x}) \approx \epsilon\boldsymbol{u}\nabla F(\boldsymbol{x})$$

Then, we can obtain:

$$\begin{aligned}
\tilde{H}(\boldsymbol{x}, \boldsymbol{u}) &= q(\boldsymbol{x} + \epsilon\boldsymbol{u}; \boldsymbol{\pi}^{(i)}) - q(\boldsymbol{x}; \boldsymbol{\pi}^{(j)}) \\
&\approx q(\boldsymbol{x}; \boldsymbol{\pi}^{(i)}) + \epsilon\boldsymbol{u}\nabla q(\boldsymbol{x}; \boldsymbol{\pi}^{(i)}) - q(\boldsymbol{x}; \boldsymbol{\pi}^{(j)}) \\
&\approx \left[ q(\boldsymbol{x}; \boldsymbol{\pi}^{(i)}) - q(\boldsymbol{x}; \boldsymbol{\pi}^{(j)}) \right] + \epsilon\boldsymbol{u}\nabla q(\boldsymbol{x}; \boldsymbol{\pi}^{(i)})
\end{aligned}$$

Thus:

$$\tilde{H}(\boldsymbol{x}, \boldsymbol{u}) - \hat{H}(\boldsymbol{x}, \boldsymbol{u}) \approx \underbrace{\left[ q(\boldsymbol{x}; \boldsymbol{\pi}^{(i)}) - q(\boldsymbol{x}; \boldsymbol{\pi}^{(j)}) \right]}_{\gamma_{i,j}} + \epsilon\boldsymbol{u} \underbrace{\left[ \nabla q(\boldsymbol{x}; \boldsymbol{\pi}^{(i)}) - \nabla F(\boldsymbol{x}) \right]}_{\zeta_i}$$

Following the proof of Theorem 3 in Qin et al. (2021), we can now obtain:

$$\begin{aligned}
P(\frac{\tilde{H}(\boldsymbol{x}, \boldsymbol{u})}{\hat{H}(\boldsymbol{x}, \boldsymbol{u})} < 0) &\leq P(|\tilde{H}(\boldsymbol{x}, \boldsymbol{u}) - \hat{H}(\boldsymbol{x}, \boldsymbol{u})| \geq |\hat{H}(\boldsymbol{x}, \boldsymbol{u})|) \\
&\leq \frac{\mathbb{E}_{\boldsymbol{\pi}}\left[ |\tilde{H}(\boldsymbol{x}, \boldsymbol{u}) - \hat{H}(\boldsymbol{x}, \boldsymbol{u})| \right]}{|\hat{H}(\boldsymbol{x}, \boldsymbol{u})|} \quad \text{according to the Markov's inequality} \\
&\leq \frac{\sqrt{\mathbb{E}_{\boldsymbol{\pi}}\left[ (\tilde{H}(\boldsymbol{x}, \boldsymbol{u}) - \hat{H}(\boldsymbol{x}, \boldsymbol{u}))^2 \right]}}{|\hat{H}(\boldsymbol{x}, \boldsymbol{u})|} \quad \text{according to the Jensen's inequality} \\
&\leq \frac{\sqrt{\mathbb{E}_{\boldsymbol{\pi}}\left[ (\gamma_{i,j} + \epsilon\boldsymbol{u}\zeta_i)^2 \right]}}{|\hat{H}(\boldsymbol{x}, \boldsymbol{u})|} \\
&\leq \frac{\sqrt{2\mathbb{E}_{\boldsymbol{\pi}}\left[ \gamma_{i,j}^2 + (\epsilon\boldsymbol{u}\zeta_i)^2 \right]}}{|\hat{H}(\boldsymbol{x}, \boldsymbol{u})|} \quad \text{according to the Cauchy's inequality}
\end{aligned}$$

∎

## C   ANALYSIS OF TRADE-OFF BETWEEN MODEL SUBSET SET SIZE AND ERROR ESTIMATION

In this section, we provide an additional analysis of the trade-off between the selection of $N$ (the subset set size) from $K$ models and the number of queries to achieve a low error estimation.

- Intuitively, a larger subset size $N$ reduces the *number* of combinations of model subsets. This results in a reduction in the number of random models presented to the attacker.

- In addition, a larger subset size $N$ also reduces the variance in estimates of gradient attempted by an attacker. Because, the prediction from a larger subset of models is more confident and the variance, for example, in output scores between these large subsets is less.

- Consequently, averaging across larger subsets of models leads to more informative responses (better gradient estimates, for example) and fewer queries to obtain low error estimations.

- In contrast, smaller $K$ values increase the uncertainty, which leads to increased variance in gradient estimation, or in other words, the difference in upper and lower bound for the gradient's value will be larger. Then following **Proposition 1**, this increases the cost of the attack, which forces the attacker to expend more queries to obtain a low error estimation of a gradient.

## D    EFFECTIVENESS OF THE PROPOSED LEARNING OBJECTIVE

The resulting *diverse* and *well-trained* models lead to the success of our proposed approach while minimizing impacts on clean accuracy. Therefore, in this section, we aim to show the effectiveness of and insights from the new training objective—sample loss—by considering models with and without sample loss.

We employ the SVGD method to train a set of models simultaneously, with and without sample loss for three tasks, MNIST (40 models), CIFAR-10 (10 models) and STL-10 (10 models). We train up to 1,000 epochs and select the best model set based on test accuracy. The results in Table 11 show that with *sample loss* objective, each individual model achieves high performance across different datasets. As a result, any randomly selected five individual models are able to obtain high accuracy (92.4%) albeit slightly lower than the accuracy achieved by the set of *All Models* (93.2%).

> In contrast, the removal of *sample loss* leads to several poor individual models (most models are below 50% accuracy). This results in an overall low performance of model randomization. For example, the random selection of five models results in a low accuracy (79%). This is not an unexpected result since the learning objective encourage model diversity while maximizing the performance of the ensemble of models. Consequently, the sample-loss objective is crucial.

Table 11: Clean accuracy under different configurations (all models, random selection of subsets of individual models and each individual model). A comparison between a set of models trained simultaneously with and without the sample loss objective on MNIST (40 models), CIFAR-10 (10 models) and STL-10 (10 models).

| Dataset | MNIST | | CIFAR-10 | | STL-10 | |
|---------|-------|-------|----------|-------|--------|-------|
| Training Objective | Without Sample Loss | With Sample Loss | Without Sample Loss | With Sample Loss | Without Sample Loss | With Sample Loss |
| All Models | 99.6% | **99.7%** | 89.8% | **93.2%** | 88.56% | **89.93%** |
| Random Selection | | | | | | |
| 8 Models | 87.3% | **99.6%** | 59.7% | **92.8%** | 82.07% | **89.11%** |
| 5 Models | 79.4% | **99.6%** | 39.8% | **92.4%** | 78.41% | **88.46%** |
| 3 Models | 69.9% | **99.6%** | 31.0% | **91.4%** | 75.13% | **87.63%** |
| Individual Model or Parameter Particle Performance | | | | | | |
| Model 1 | 50.7% | **99.3%** | 15.1% | **88.5%** | 58.2% | **83.03%** |
| Model 2 | 36.6% | **99.5%** | 13.8% | **88.3%** | 58.69% | **81.44%** |
| Model 3 | 22.8% | **99.3%** | 13.5% | **88.5%** | 51.51% | **82.25%** |
| Model 4 | 42.2% | **99.2%** | 10.0% | **88.1%** | 51.7% | **84.79%** |
| Model 5 | 32.7% | **99.5%** | 9.3% | **88.9%** | 55.11% | **82.7%** |
| Model 6 | 35.4% | **99.4%** | 12.4% | **86.9%** | 60.39% | **84.88%** |
| Model 7 | 35.6% | **99.4%** | 11.3% | **88.4%** | 52.94% | **83.65%** |
| Model 8 | 32.0% | **99.4%** | 12.4% | **89.7%** | 43.85% | **83.79%** |
| Model 9 | 55.6% | **99.3%** | 10.2% | **87.7%** | 52.79% | **80.6%** |
| Model 10 | **99.3%** | **99.3%** | 80.8% | **88.4%** | 71.08% | **82.78%** |

**Analysis.** As discussed in Section 3.6.2, we incorporate sample loss in the joint training objective as a new constraint to encourage individual model learning while training a diverse set of models. This helps individual models obtain high performance and prevents a subset of models from collapsing. For instance, without *sample loss* models 2-8 on MNIST, models 1-9 on CIFAR-10 are completely poor as shown in Table 11. Because:

- Minimizing the loss over an average of logits for a subset of model faces the same problem we tried to address with the introduction of our learning objective (Sample Loss)—minimizing the loss over an average of logits for a subset of models promotes strong ensemble performance but does not guarantee that *each* individual model will perform well.

- Individual model performance, as we mentioned in Section 3.6.2, is very important to ensure minimal performance degradation for our defense. Because we want any randomly selected model or set to be well-performing.

- To this end, the proposed objective, through the joint training process, promotes diversity among models and ensures each individual model maintains strong performance.

# E  OVERHEAD ANALYSIS

Our approach provides significant improvements in robustness. However, achieving robustness requires training (a one-time cost) and model storage cost. In this section, we analyze these costs and investigate a method for mitigating the cost. Followed by an experimental evaluation of the method with the high resolution `ImageNet` task.

Cost and complexity comparisons shown in Tables 12, 13 and 14 for training a single model versus sets of models as used in our experiments show that achieving better robustness does come at some cost.

Table 12: Training and inference times of different models between different defense mechanisms RND, RF and Disco(SVGD+) (40 models for `MNIST`, 10 models for `CIFAR-10/STL-10`). Here, for fairness, we assume the Disco methods process inputs one model at a time (*sequential*) and *five* models are randomly sampled. In practice, the inference times can be similar to a single model as the forward pass of the input can occur in parallel across an ensemble.

| Datasets | Training Time | | Inference Time | | | |
|---|---|---|---|---|---|---|
| | Single Model (RND and RF) | A set of models (Disco) | Undefended | RND | RF | Disco (*sequential*) |
| MNIST | ∼0.5 hr | ∼12.5 hrs | 0.7 $\mu$s | 1.14 $\mu$s | 1.53 $\mu$s | 3.8 $\mu$s |
| CIFAR-10 | ∼1.5 hr | ∼72 hrs | 1.9 $\mu$s | 2.61 $\mu$s | 2.92 $\mu$s | 9.8 $\mu$s |
| STL-10 | ∼1.2 hr | ∼60 hrs | 2.5 $\mu$s | 3.12 $\mu$s | 3.48 $\mu$s | 12.8 $\mu$s |

Table 13: Inference times of single model and DISCO (with no parallelisation of models) on `ImageNet`.

| Datasets | OpenCLIP | DISCO (subset of two out of five) |
|---|---|---|
| ImageNet | 4.4 ms | 9ms |

Table 14: Trainable Parameters and Storage Consumption of models trained on different datasets between a single model and a set of models (Disco(SVGD+))—40 models for `MNIST`, 10 models for `CIFAR-10/STL-10`.

| Datasets | Trainable Parameters | | Storage Consumption | |
|---|---|---|---|---|
| | Single Model (RND and RF) | A set of models (Disco) | Single Model (RND and RF) | A set of models (Disco) |
| MNIST | 0.312 M | 12.5 M | 1.19 MB | 47.7 MB |
| CIFAR-10 | 14.73 M | 147.3 M | 56.18 MB | 561.84 MB |
| STL-10 | 11.18 M | 111.8 M | 43.12 MB | 426.55 MB |

# F  AMELIORATING OVERHEAD FOR LARGE-SCALE TASKS

In general, the use of multiple models does lead to increasing the training and storage burden. RND and RF use a single model, whereas we employ a set of $n$ models, so the number of parameters in our approach is $n\times$ higher, and the memory consumption is also larger. In practical applications, the cost can be mitigated:

- Recent work research in the area of model tuning with low-rank adapters (LoRAs) Hu et al. (2021) can mitigate the cost of building large-scale practical ensembles.

- The study in Doan et al. (2024) develops a method for a pre-trained model to be tuned with only a 1% increase in trainable parameters and storage costs to build ensembles of diverse models using SVGD. The authors employ the pre-trained OpenCLIP Radford et al. (2021) model for the `ImageNet` task and LlaVA for a visual question and answer task.

Next, we adopt the method of model fine tuning to demonstrate how Disco(), the model randomization method, can be implemented for *practical tasks* `ImageNet`, and *for a practical, large-scale model* OpenCLIP.

### F.1 EFFECTIVENESS ON HIGH-RESOLUTION LARGE-SCALE DATASET AND THE PRACTICAL LARGE-SCALE **OPENCLIP** MODEL

In this section, we demonstrate the effectiveness of our method on high-resolution datasets like `ImageNet` and with a large-scale, piratical model, *OpenCLIP* Radford et al. (2021).

**Robustness Comparison**

Inspired by recent work Doan et al. (2024), we adopt the technique of fine-tuning pre-trained models to obtain well-trained, large-scale models at a fraction of the cost of training an ensemble from initialization. The authors employ SVGD to achieve model diversity and use low-rank adapters to significantly reduce the cost of building the ensemble to better approximate a multi-modal Bayesian posterior.

Table 15: $l_\infty$ **objective**. Robustness (higher $\uparrow$ is stronger) of defenses against SQUAREATTACK on the `ImageNet` with an OpenCLIP.

| Methods | $l_\infty$ =0.025 | 0.05 | 0.075 | 0.1 |
|---------|------------------|------|-------|-----|
| RND | 83.39% | 61.95% | 43.37% | 24.89% |
| RF | 86.45% | 65.1 % | 51.14% | 35.83% |
| **Disco** | **90.76%** | **72.51%** | **56.17%** | **45.4%** |

As a demonstration of a practical application with a large-scale model, we also use the pre-trained, OpenCLIP, large-scale model with low-rank adaptors (LoRA) Hu et al. (2021) as in Doan et al. (2024) to build a sample of five models for the `ImageNet` task. The ensemble achieves approximately 78% clean accuracy on the test set. We used a random selection of two out of five models in our method for the defense, where the clean accuracy of two out of 5 models is approximately 77%. For RND and RF defenses, we fine-tune the CLIP model for the `ImageNet` task to achieve 76.07 % clean accuracy and, for a fair comparison, we choose hyperparameters such that the clean accuracy drop, after injecting noise, is approximately 1%. In this experiment, we randomly select 100 correctly classified images from the `ImageNet` test set and use the SQUAREATTACK ($l_\infty$) against the models.

The results in Table 15 show that our approach achieves the best results across various perturbation budgets on `ImageNet` task compared to both RND and RF methods. Importantly, our approach is able to achieve up to 9.6% increase in robustness above the next best performing method RF with an ensemble of just 5 models. We also show the clean accuracy on `ImageNet` for reference.

Table 16: Clean accuracy and accuracy drop ($\downarrow \Delta$) achieved by undefended and defended models on `ImageNet`.

| Dataset | Single Model | RND ($\downarrow \Delta$) | RF ($\downarrow \Delta$) | Disco ($\downarrow \Delta$) |
|---------|-------------|---------------------------|--------------------------|------------------------------|
| `ImageNet` | 76.07% | 75.04% ($\downarrow 1.03\%$) | 75.13% ($\downarrow 0.94\%$) | 76.96% ($\uparrow 0.89\%$) |

**Cost Mitigation Analysis**

The results in Table 17 show that our approach can be realized in large-scale network architecture like OpenCLIP and yet achieve the best robustness results across various perturbation budgets on the `ImageNet` task compare to both RND and RF methods. Now, only a marginal cost increase is needed to achieve significant improvements in robustness.

**Summary**

What we propose are *marginal* cost increases in terms of model parameter and storage to achieve significant improvements in robustness.

Table 17: Trainable Parameters and Storage Consumption of five OpenCLIP with LoRA trained on `ImageNet`. Notably, we begin with a single *pre-trained* OpenCLIP model and subsequently construct the ensemble of five models while tuning the model for the `ImageNet`.

| Models | Single CLIP | A set of CLIPs with LoRA |
|---|---|---|
| Trainable Parameters | 114 M | 1.84 M (↑1.6%, 0.32% per model) |
| Storage Consumption | 433 MB | 439 MB (↑1.38%, 0.28% per model) |

- Effectively, we demonstrate <1.6% increase in overhead can yield up to 9.6% better robustness (when compared to the next best defense method) on a large-scale network of practical significance.

- Now, adding a model incurs <0.32% overhead in terms of trainable parameters or storage.

Overall, these results also demonstrate that our model randomization method is:

- Practical for implementation

- Effective across different datasets and model types.

# G EFFECTIVENESS AGAINST A STRONGER ATTACK IN A SURROGATE MODEL SETTING

In this section, we further assess the robustness of the different defense mechanisms against the state-of-the-art attack using a surrogate model. In the Prior-Bayesian Optimization (P-BO) attack Cheng et al. (2024), it integrates transfer-based and query-based techniques. The results in Table 18 demonstrate that our defense outperforms RND and RF and effectively fortifies against the strong P-BO attack setting. This underpins the capability of `Disco` to withstand even the most advanced query-based attacks. This reinforces the strength and general applicability of our approach in defending against cutting-edge black-box attack methods such as P-BO.

Table 18: $l_\infty$ **objective**. Robustness (higher ↑ is stronger) of defenses against P-BO with `CIFAR-10` task.

| Methods | $l_\infty$ =0.02 | 0.04 | 0.06 | 0.08 | 0.1 |
|---|---|---|---|---|---|
| Single (*undef*) | 0.0% | 0.0% | 0.0% | 0.0% | 0.0% |
| RND | 70.33% | 31.47% | 15.75% | 7.23% | 6.65% |
| RF | 66.43% | 28.04 % | 13.67% | 8.34% | 6.21% |
| **Disco** | **79.98**% | **47.94**% | **29.8**% | **18.08**% | **12.16**% |

# H ALTERNATIVE APPROACHES FOR MODEL DIVERSITY PROMOTION

## H.1 FORMULATION OF TRAINING OBJECTIVES

**Ensembles employing Random Initialization Approaches.** Lakshminarayanan et al. (2017) proposed to train a set of models—*Ensemble*—with random initializations independently. This training is formulated as follows:

$$\min_{\boldsymbol{\theta}_k} \quad \mathbb{E}_{(\boldsymbol{x},y)\sim\mathcal{D}}\Big[\ell(f_k(\boldsymbol{x};\boldsymbol{\theta}_k), y)\Big], \tag{13}$$

where $\boldsymbol{\theta}_i$ denotes the weights of the $i$-th model, and $\ell(.,.)$ is the loss (*i.e.* cross-entropy).

**Gradient-based Approach.** Teney et al. (2022) introduced a method encouraging diversity over a set of models by quantifying the similarity of the gradient of the top predicted score of each model with respect to its features. This method aims to train a model set to discover predictive patterns commonly missed by learning algorithms and promote diversity across the model set. In our study, we adopt their *Diversity Regularizer* (DivReg) to encourage the model diversity and formulate the

training objective as follows:

$$\min_{\boldsymbol{\Theta}} \mathbb{E}_{(\boldsymbol{x},y)\sim\mathcal{D}} \Big[ \sum_{k=1}^{K} \ell(f_k(\boldsymbol{x};\boldsymbol{\theta}_k), y) + \lambda_{\text{reg}} \sum_{i\neq j} \delta_{f_i,f_j} \Big], \tag{14}$$

where $\delta_{f_i,f_j} = \nabla_h f_i(\boldsymbol{h}_i) \cdot \nabla_h f_j(\boldsymbol{h}_j)$, $\lambda_{\text{reg}}$ controls the strength of the regularizer, $\nabla_h f_i$ and $\nabla_h f_j$ denote the gradient of the top predicted score of models $f_i$ and $f_j$ w.r.t their own features $\boldsymbol{h}_i$ and $\boldsymbol{h}_j$.

**Score-based Approach.** Lee et al. (2023) proposed an approach to training a collection of diverse models by independently training each head pair to make predictions. In our study, we adopt their loss function to encourage model diversity. The training objective is formulated as follows:

$$\min_{\boldsymbol{\Theta}} \mathbb{E}_{(\boldsymbol{x},y)\sim\mathcal{D}} \Big[ \sum_{k=1}^{K} \ell(f_k(\boldsymbol{x};\boldsymbol{\theta}_k), y) + \lambda_{\text{MI}} \sum_{k\neq i} \mathcal{L}_{\text{MI}}(f_k(\boldsymbol{x};\boldsymbol{\theta}_k), f_i(\boldsymbol{x};\boldsymbol{\theta}_i)) \Big], \tag{15}$$

where $\mathcal{L}_{\text{MI}}(f(\boldsymbol{x};\boldsymbol{\theta}_k), f(\boldsymbol{x};\boldsymbol{\theta}_i)) = D_{KL}(p(\hat{y}_k, \hat{y}_i) \parallel p(\hat{y}_k) \otimes p(\hat{y}_i))$, $D_{KL}(. \parallel .)$ is the KL divergence and $\hat{y}_i$ is the predicted label from $f_i(\boldsymbol{x};\boldsymbol{\theta}_i)$, $\lambda_{\text{MI}}$ controls the strength of mutual information loss $\mathcal{L}_{\text{MI}}$, $p(\hat{y}_k, \hat{y}_i)$ is the empirical estimate of the joint distribution and $p(\hat{y}_k)$, $p(\hat{y}_i)$ are the empirical estimates of the marginal distributions.

## H.2 Model Diversity Analysis

As we discussed in Section 3.6, more diversity among individual models or particles results in enhanced output diversity. Therefore, we use Jensen–Shannon divergence as a metric to illustrate model diversity promoted by different methods. We measure diversity by calculating the Jensen–Shannon divergence between the average softmax outputs of all models $\hat{p}(x_i) = \frac{1}{K} \sum_{k=1}^{K} \text{softmax}[f(\boldsymbol{x}_i;\boldsymbol{\theta}_k)]$ versus the softmax output of each individual model $p_k(x_i) = \text{softmax}[f(\boldsymbol{x}_i;\boldsymbol{\theta}_k)]$ and then averaging across all samples from a test set as follows:

$$D_{JS}^{(k)} = \frac{1}{2} \sum_{i}^{N} \Big( D_{KL}\Big(\hat{p}(x_i) \parallel \frac{\hat{p}(x_i) + p_k(x_i)}{2}\Big) + D_{KL}\Big(p_k(x_i) \parallel \frac{\hat{p}(x_i) + p_k(x_i)}{2}\Big) \Big),$$

where $D_{\text{KL}}(. \parallel .)$ is the KL divergence, k represent the individual model $k$-th and $N$ denotes the size of a dataset. The results as shown in Section 4.4 and Figure 3 demonstrate that our proposed method (SVGD+) is able to achieve greater diversity among individual parameter particles. *These empirical findings support the assertions of our hypotheses in Section 3 and the robustness of the defense method we formulated.*

## H.3 Diversity Analysis of Individual Models Learned With Different Training Objectives

Similar to Section 4.1, we measure the diversity between every pair of individual models which can be computed with Equation 16. The results demonstrated in Figure 4 show highly diverse nature of

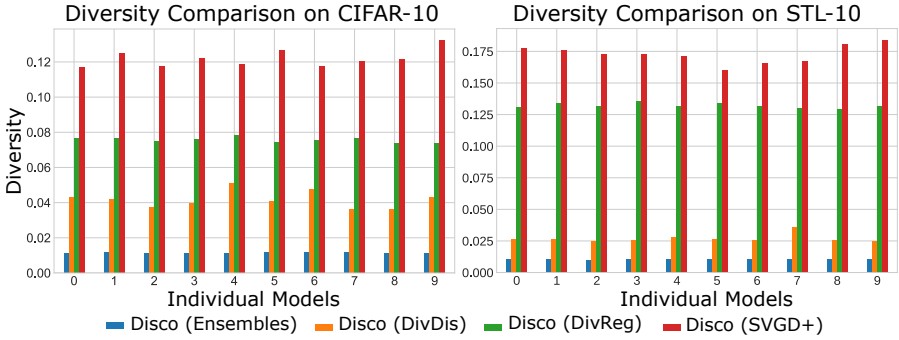

Figure 3: Model diversity using JS divergence among Ensemble, DivDis, DivReg and SVGD+ (**ours**).

the models (larger range of colors for model vs. model results) and that each individual model trained by our proposed approach obtains higher diversity (denoted by lighter colors).

$$D_{JS}^{(k,j)} = \frac{1}{2} \sum_{i}^{N} \Big( D_{KL}\Big( p_k(x_i) \parallel \frac{p_k(x_i) + p_j(x_i)}{2} \Big) + D_{KL}\Big( p_j(x_i) \parallel \frac{p_k(x_i) + p_j(x_i)}{2} \Big) \Big), \quad (16)$$

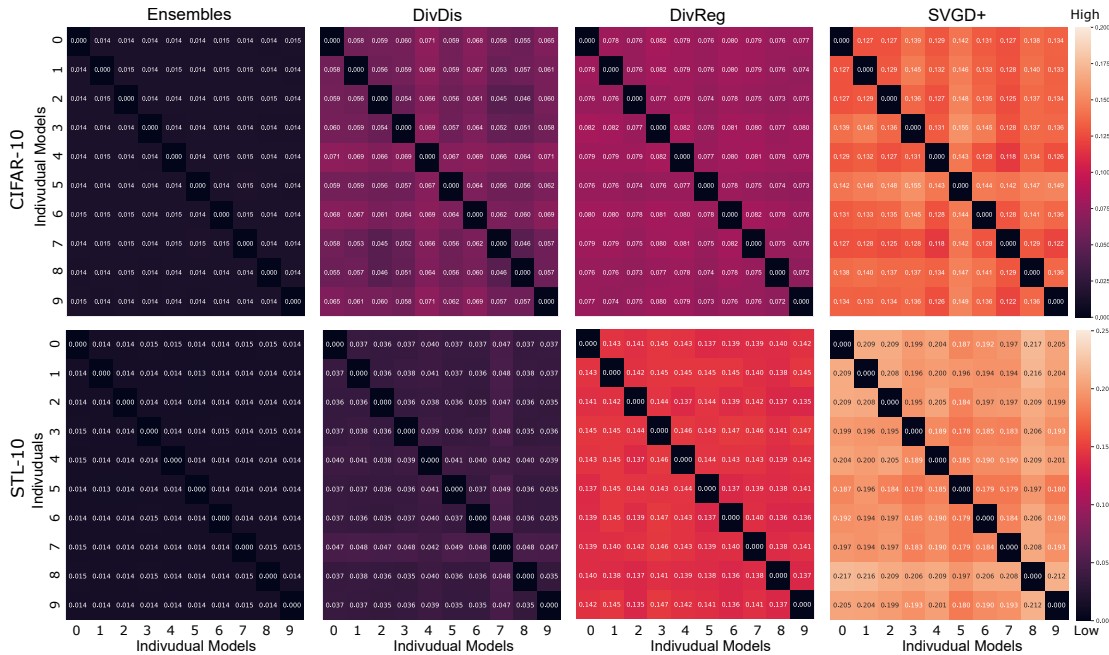

Figure 4: The diversity measurement (using Jensen–Shannon divergence) between every pair of individual models trained by Ensemble, DivDis, DivReg and our proposed method SVGD+ (Ours) on CIFAR-10 and STL-10.

## I ROBUSTNESS ACHIEVED WITH DISCO USING ALTERNATIVE DIVERSITY PROMOTION APPROACHES

Table 19: $l_2$ **objective attacks, CIFAR-10.** Robustness (higher ↑ is stronger) of diversity promotion approaches against SIGNHUNTER with the CIFAR-10 task.

| Methods | $l_2 = 0.8$ | 1.6 | 2.4 | 3.2 | 4.0 |
|---|---|---|---|---|---|
| Disco(Ensemble) | 99.69% | 97.72% | 93.49% | 89.22% | 81.55% |
| Disco(DivDis) | 99.87% | 98.74 % | 95.32% | 91.97% | 85.6% |
| Disco(DivReg) | 99.96% | 99.07% | 96.38% | 91.42% | 86.95% |
| **Disco(SVGD+)** | **99.96%** | **99.25%** | **97.61%** | **93.63%** | **90.24%** |

In this section, we conduct extensive experiments to evaluate the robustness of alternative approaches for model diversity promotion (Ensemble, DivDis and DivReg) together with our proposal (SVGD+) against score-based adversarial attack NESATTACK ($l_\infty$), SIGNHUNTER ($l_2$, $l_\infty$), SQUAREAT-TACK ($l_\infty$), SPARSERS ($l_0$) and decision-based attacks HOPSKIPJUMP ($l_2$) and SPAEVO ($l_0$) on CIFAR-10. The results in Tables 20, 19, 21 and 22 demonstrate that our proposed defense mechanism outperforms all other alternative approaches.

## J COMPARISON WITH ADAPTIVE DIVERSITY PROMOTING (ADP) METHOD

Adaptive diversity promoting (ADP) Pang et al. (2019) method employs a regularizer while training an ensemble to encourage model diversity. This results in enhancing the robustness for the ensemble because it is difficult to transfer adversarial examples among individual models. Here, we investigate

Table 20: **CIFAR-10**. Robustness (higher ↑ is stronger) of different defense methods against NESATTACK, SIGNHUNTER and SQUAREATTACK attacks under the $l_\infty$ perturbation objective.

| Attack | Methods | $l_\infty$ =0.02 | 0.04 | 0.06 | 0.08 | 0.1 |
|---|---|---|---|---|---|---|
| NESATTACK | Disco(Ensemble) | 98.82% | 95.32 % | 89.77% | 85.43% | 81.11% |
| | Disco(DivDis) | 99.51% | **98.5 %** | **95.05%** | **92.64%** | **88.54%** |
| | Disco(DivReg) | 99.45% | 97.54 % | 92.94% | 89.53% | 84.65% |
| | **Disco(SVGD+)** | 99.7% | 97.93% | 94.39% | 90.5% | 86.77% |
| SIGNHUNTER | Disco(Ensemble) | 99.88% | 96.95 % | 90.03% | 81.75% | 73.79% |
| | Disco(DivDis) | 99.88% | 98.11 % | 92.83% | 84.97% | 77.44% |
| | Disco(DivReg) | 99.98% | 98.30 % | 90.13% | 77.4% | 64.86% |
| | **Disco(SVGD+)** | 99.97% | **98.95%** | **95.56%** | **90.7%** | **84.22%** |
| SQUAREATTACK | Disco(Ensemble) | 99.77% | 91.08 % | 71.69% | 52.67% | 39.69% |
| | Disco(DivDis) | 99.96% | 96.85 % | 85.23% | 67.96% | 52.29% |
| | Disco(DivReg) | **99.99%** | 96.06 % | 83.1% | 64.1% | 48.63% |
| | **Disco(SVGD+)** | 99.97% | **96.91%** | **86.52%** | **70.22%** | **55.77%** |

Table 21: **CIFAR-10**. Robustness (higher ↑ is stronger) of different model diversity promotion schemes against SPARSERS ($l_0$).

| Methods | $l_0$ =16px | 32px | 48px | 64px | 80px |
|---|---|---|---|---|---|
| Disco(Ensemble) | 50.12% | 36.82% | 25.97% | 23.16% | 19.06% |
| Disco(DivDis) | 57.94% | 41.2% | 35.15% | 29.75% | 25.59% |
| Disco(DivReg) | 61.38% | 44.95% | 39.17% | 32.62% | 28.0% |
| **Disco(SVGD+)** | **63.85%** | **47.84%** | **41.59%** | **36.81%** | **31.24%** |

Table 22: **Decision-based, CIFAR-10**. Robustness (higher ↑ is stronger) of different model diversity promotion schemes against HOPSKIPJUMP ($l_2$) and SPAEVO ($l_0$).

| Methods | HOPSKIPJUMP | | | | | SPAEVO | | | | |
|---|---|---|---|---|---|---|---|---|---|---|
| | $l_2$ =0.8 | 1.6 | 2.4 | 3.2 | 4.0 | $l_0$ =4px | 8px | 12px | 16px | 20px |
| Disco(Ensemble) | 99.82% | 99.12 % | 97.94% | 96.95% | 96.31% | 92.07% | 91.79 % | 91.69% | 91.67% | 91.66% |
| Disco(DivDis) | 99.94% | **99.59 %** | **98.94%** | 98.04% | 97.33% | 93.68% | 93.3% | 93.1% | 93.07% | 92.97% |
| Disco(DivReg) | 99.98% | 99.25 % | 98.26% | 97.39% | 96.41% | 93.9% | 93.53% | 93.42% | 93.35% | 93.32% |
| **Disco(SVGD+)** | **99.94%** | 99.4% | 98.77% | **98.04%** | **97.43%** | **96.17%** | **95.99%** | **95.94%** | **95.88%** | **95.84%** |

the performance provided by the model diversification method against query-based black-box attacks under our proposed model randomization method and compare it with our proposed SVGD+ method for building a set of diverse and well-performing models.

Table 23: **CIFAR-10**. Compare the robustness (higher ↑ is stronger) of our approach using SVGD+ versus ADP against SQUAREATTACK ($l_\infty$) and SIGNHUNTER ($l_\infty$). We randomly select a subset of five models (from ten models).

| Groups | Methods | $l_\infty$ =0.02 | 0.04 | 0.06 | 0.08 | 0.1 |
|---|---|---|---|---|---|---|
| SIGNHUNTER ($l_\infty$) | Disco(ADP) | **99.99%** | **99.59%** | **96.01%** | 89.78% | 82.19% |
| | **Disco(SVGD+)** | 99.97% | 99.95% | 95.56% | **90.71%** | **84.22%** |
| SQUAREATTACK ($\infty$) | Disco(ADP) | **99.99%** | 96.31% | 83.99% | 65.28% | 50.41% |
| | **Disco(SVGD+)** | 99.97% | **96.91%** | **86.52%** | **70.22%** | **55.77%** |

Table 24: **CIFAR-10**. Compare the robustness (higher ↑ is stronger) of our approach using SVGD+ versus ADP against (SQUAREATTACK ($l_2$), SIGNHUNTER ($l_2$)). We randomly select a subset of five models (from ten models).

| Groups | Methods | $l_2$ =0.8 | 1.6 | 2.4 | 3.2 | 4.0 |
|---|---|---|---|---|---|---|
| SIGNHUNTER ($l_2$) | Disco(ADP) | **99.98%** | **99.62 %** | 97.54% | **94.06%** | 89.07% |
| | **Disco(SVGD+)** | 99.96% | 99.24 % | **97.61%** | 93.63% | **90.24%** |
| SQUAREATTACK ($2$) | Disco(ADP) | **99.91%** | **97.12 %** | **87.71%** | 76.35% | 63.81% |
| | **Disco(SVGD+)** | 99.56% | 95.62% | 87.07% | **76.50%** | **65.76%** |

We conduct extensive experiments to compare the robustness of our approach using SVGD+ and ADP under our framework with a configuration of random five out of 10 models against black-box attacks, SQUAREATTACK ($l_2$, $l_\infty$) and SIGNHUNTER ($l_2$, $l_\infty$). The results in Tables 23 and 24 show that our proposed model randomization performs well with the learning objective introduced in Pang

Table 25: Clean accuracy and clean accuracy drop ($\downarrow \Delta$) comparison between ADP models versus SVGD+ training objective based models on `CIFAR-10`. *All* represents results from the entire ensemble of models while Disco(.) represents performance under the model randomization configured with five out of 10 models.

| Single Model | ADP (All) | Disco(ADP) ($\downarrow \Delta$) | SVGD+ (All) | Disco(SVGD+) ($\downarrow \Delta$) |
|---|---|---|---|---|
| 92.09 | 94.56% | 93.29% (↑**1.29%**) | 93.19% | 92.26% (↑**0.17%**) |

et al. (2019). The results also demonstrate that ADP can encourage diversity and Disco(ADP) can achieve comparable robustness to Disco(SVGD+) under *low* perturbation budgets. Under increasing perturbations, models learned with SVGD+ demonstrates improved robustness.

Interestingly, from the analysis of clean accuracy drop in Table 25, we can observe the learning objective we introduced in Section 3.6.2 allows Disco(SVGD+) to achieve a lower clean accuracy drop than Disco(ADP).

## K    EVALUATIONS WITH DIFFERENT RANDOMIZED MODEL SELECTION STRATEGIES

**On MNIST.** In this section, we provide additional results for training a set of 10, 20 and 40 models using Ensemble, DivDis, DivReg and our proposed method under SQUAREATTACK ($l_2$). For robustness evaluation and comparison, we chose different settings with different sizes of model subsets. For instance, we sample 1, 3, 5, 8 of 10 models and sample 1, 3, 5, 10 of 20 models. For 40 models, we sample 1, 3, 5, 20 and 30 of 40 models. The results in Tables 26, 27, 28 and 29 provide further evidence to demonstrate that our proposed method is more robust than other diversity promotion methods across different distortions and settings[2].

Table 26: $l_2$ **objective.** Robustness (higher ↑ is stronger) of different defense methods against SIGNHUNTER and SQUAREATTACK.

| Methods | SIGNHUNTER | | | | | SQUAREATTACK | | | | |
|---|---|---|---|---|---|---|---|---|---|---|
| | $l_2$=0.8 | 1.6 | 2.4 | 3.2 | 4.0 | $l_2$=0.8 | 1.6 | 2.4 | 3.2 | 4.0 |
| Single (*undef*) | 96.8% | 56.6% | 11.0% | 2.0% | 0.0% | 81.4% | 7.0% | 0.0% | 0.0% | 0.0% |
| Ensemble (*undef*) | 98.6% | 93.2% | 56.6% | 13.8% | 3.0% | 95.2% | 38.8% | 0.8% | 0.0% | 0.0% |
| RND | 99.76% | 94.19% | 78.04% | 63.03% | 49.47% | 99.42% | 92.95% | 77.31% | 60.59% | 45.12% |
| RF | 99.98% | 99.68 % | 96.95% | 86.69% | 70.48% | 99.99% | 99.59% | 95.19% | 83.54% | 68.79% |
| **Disco** | **100%** | **99.99%** | **99.78%** | **99.78%** | **99.25%** | **100%** | **99.76%** | **97.46%** | **88.98%** | **77.75%** |

Table 27: **MNIST**. A robustness comparison (higher ↑ is stronger) between our proposed method and other methods against SQUAREATTACK. For the evaluation of different diversity-promotion methods, we train a set of 10 models and randomly select a subset of a different number of models.

| Random | Methods | $l_2$=0.8 | 1.6 | 2.4 | 3.2 | 4.0 |
|---|---|---|---|---|---|---|
| 1 | Disco(Ensemble) | 98.4% | 91.9% | 87.1% | 82.0% | 74.6% |
| | Disco(DivDis) | 98.3% | 92.9% | 86.6% | 78.9% | 70.9% |
| | Disco(DivReg) | 96.1% | 88.5% | 80.9% | 72.5% | 66.5% |
| | **Disco(SVGD+)** | **98.7%** | **94.4%** | **89.3%** | **86.0%** | **80.0%** |
| 3 | Disco(Ensemble) | 99.9% | 98.7% | 91.6% | 78.7% | 68.0% |
| | Disco(DivDis) | 99.8% | 97.6% | 87.8% | 76.9% | 62.5% |
| | Disco(DivReg) | 99.8% | 85.5% | 88.1% | 77.6% | 69.1% |
| | **Disco(SVGD+)** | **99.9%** | **99.6%** | **95.4%** | **83.3%** | **70.3%** |
| 5 | Disco(Ensemble) | 100% | 98.6% | 90.5% | 74.4% | 55.8% |
| | Disco(DivDis) | 99.8% | 96.3% | 85.0% | 71.6% | 57.0% |
| | Disco(DivReg) | 99.9% | 94.4% | 83.8% | 72.1% | 60.0% |
| | **Disco(SVGD+)** | **100%** | **99.4%** | **95.3%** | **81.1%** | **63.8%** |
| 8 | Disco(Ensemble) | 100% | 98.5% | 90.3% | 73.5% | 54.2% |
| | Disco(DivDis) | 99.5% | 94.3% | 82.6% | 67.1% | 53.5% |
| | Disco(DivReg) | 99.9% | 93.2% | 80.7% | 72.3% | 59.7% |
| | **Disco(SVGD+)** | **99.99%** | **99.24%** | **91.35%** | **76.48%** | **59.43%** |

**On CIFAR-10.** In this section, we provide additional results for robustness evaluation and comparison in different settings with different sizes of model subsets (*i.e.* 1, 3, 5, and 8) under SQUAREATTACK ($l_2$). The results in Table 30 show that our proposed method is more robust than other diversity promotion methods across different distortions and settings.

---

[2]Please see Table 31 for clean accuracy results under different model training and random selection strategies

Table 28: **MNIST**. A robustness comparison (higher ↑ is stronger) between our proposed method and other methods against SQUAREATTACK. For the evaluation of different diversity-promotion methods, we train a set of 20 models and randomly select a subset of a different number of models.

| Random | Methods | $l_2 = 0.8$ | 1.6 | 2.4 | 3.2 | 4.0 |
|---|---|---|---|---|---|---|
| 1 | Disco(Ensemble) | 99.2% | 96.1% | 91.6% | 85.1% | 75.4% |
| | Disco(DivDis) | 99.0% | 95.5% | 91.1% | 82.7% | 74.5% |
| | Disco(DivReg) | 96.5% | 91.8% | 83.7% | 76.8% | 68.6% |
| | **Disco(SVGD+)** | **99.4%** | **97.3%** | **94.2%** | **90.2%** | **83.5%** |
| 3 | Disco(Ensemble) | 100% | 99.5% | 95.4% | 85.7% | 70.8% |
| | Disco(DivDis) | 100% | 97.8% | 89.9% | 79.3% | 66.5% |
| | Disco(DivReg) | 100% | 97.0% | 91.1% | 83.9% | 76.5% |
| | **Disco(SVGD+)** | **100%** | **99.8%** | **98.1%** | **90.5%** | **78.8%** |
| 5 | Disco(Ensemble) | 100% | 99.3% | 93.2% | 77.5% | 62.8% |
| | Disco(DivDis) | 99.8% | 97.5% | 90.6% | 76.8% | 60.4% |
| | Disco(DivReg) | 99.9% | 97.8% | 90.6% | 76.8% | 60.4% |
| | **Disco(SVGD+)** | **100%** | **99.4%** | **94.8%** | **85.1%** | **70.0%** |
| 10 | Disco(Ensemble) | 99.9% | 98.2% | 86.5% | 67.1% | 46.0% |
| | Disco(DivDis) | 99.7% | 93.1% | 79.1% | 65.0% | 50.1% |
| | Disco(DivReg) | 99.9% | 94.0% | 83.0% | 72.9% | **62.6%** |
| | **Disco(SVGD+)** | **100%** | **99.5%** | **95.0%** | **76.8%** | 60.0% |

Table 29: **MNIST**. A comparison of robustness (higher ↑ is better) between Disco(SVGD+) and other learning methods against SQUAREATTACK. For the evaluation of different diversity promotion methods, we train a set of 40 models and randomly select a subset of a different number of models.

| Random | Methods | $l_2 = 0.8$ | 1.6 | 2.4 | 3.2 | 4.0 |
|---|---|---|---|---|---|---|
| 1 | Disco(Ensemble) | 99.6% | 97.3% | 93.4% | 89.0% | 80.2% |
| | Disco(DivDis) | 99.6% | 97.4% | 93.9% | 88.1% | 82.6% |
| | Disco(DivReg) | 99.2% | 96.2% | 91.8% | 84.3% | 77.4% |
| | **Disco(SVGD+)** | **99.7%** | **98.9%** | **97.2%** | **93.5%** | **88.2%** |
| 3 | Disco(Ensemble) | 100% | 99.4% | 94.2% | 85.2% | 74.6% |
| | Disco(DivDis) | 100% | 98.6% | 93.8% | 83.7% | 73.1% |
| | Disco(DivReg) | 100% | 99.0% | 93.0% | 79.7% | 67.6% |
| | **Disco(SVGD+)** | **100%** | **99.8%** | **98.0%** | **91.4%** | **77.8%** |
| 5 | Disco(Ensemble) | 100% | 99.5% | 95.8% | 84.9% | 70.8% |
| | Disco(DivDis) | 100% | 98.6% | 94.3% | 79.9% | 68.9% |
| | Disco(DivReg) | 100% | 98.4% | 90.5% | 79.5% | 67.9% |
| | **Disco(SVGD+)** | **100%** | **99.8%** | **97.9%** | **90.1%** | **76.5%** |
| 20 | Disco(Ensemble) | 100% | 97.6% | 86.4% | 68.0% | 49.0% |
| | Disco(DivDis) | 99.8% | 95.9% | 85.5% | 72.2% | 54.8% |
| | Disco(DivReg) | 99.7% | 96.1% | 83.3% | 67.0% | 51.3% |
| | **Disco(SVGD+)** | **100%** | **99.3%** | **94.4%** | **77.5%** | **56.8%** |
| 30 | Disco(Ensemble) | 99.9% | 96.8% | 81.2% | 60.6% | 40.0% |
| | Disco(DivDis) | 99.9% | 95.9% | 80.0% | 64.9% | 46.9% |
| | Disco(DivReg) | 99.5% | 93.9% | 77.3% | 59.7% | 43.2% |
| | **Disco(SVGD+)** | **100%** | **98.6%** | **91.9%** | **70.4%** | **52.2%** |

Table 30: **CIFAR-10**. A robustness comparison (higher ↑ is stronger) between our approach and other methods against SQUAREATTACK. For the evaluation of different diversity promotion methods, we train a set of 10 models and randomly select a subset of a different number of models.

| Random | Methods | $l_2 = 0.8$ | 1.6 | 2.4 | 3.2 | 4.0 |
|---|---|---|---|---|---|---|
| 1 | Disco(Ensemble) | 90.0% | 83.6% | 75.4% | 64.2% | 55.1% |
| | Disco(DivDis) | **95.1%** | **90.1%** | **82.6%** | 72.0% | 59.1% |
| | Disco(DivReg) | 90.6% | 86.2% | 79.5% | 69.6% | 59.5% |
| | **Disco(SVGD+)** | 90.2% | 86.9% | 82.2% | **75.2%** | **67.6%** |
| 3 | Disco(Ensemble) | 97.1% | 88.3% | 78.6% | 67.4% | 55.4% |
| | Disco(DivDis) | 99.2% | 96.1% | 86.2% | 75.8.3% | 62.1% |
| | Disco(DivReg) | 99.6% | 93.5% | 84.3% | 72.1% | 60.3% |
| | **Disco(SVGD+)** | **99.8%** | **96.7%** | **90.0%** | **82.2%** | **72.6%** |
| 5 | Disco(Ensemble) | 97.7% | 89.0% | 76.1% | 63.1% | 52.3% |
| | Disco(DivDis) | 99.0% | 93.9% | 83.0% | 70.8% | 55.7% |
| | Disco(DivReg) | 99.0% | 91.6% | 78.5% | 65.3% | 53.6% |
| | **Disco(SVGD+)** | **99.6%** | **95.6%** | **87.1%** | **76.5%** | **65.8%** |
| 8 | Disco(Ensemble) | 98.2% | 87.9% | 76.9% | 63.5% | 52.2% |
| | Disco(DivDis) | 99.1% | 94.1% | 82.7% | 70.4% | 56.4% |
| | Disco(DivReg) | 99.2% | 90.9% | 76.3% | 64.6% | 51.6% |
| | **Disco(SVGD+)** | **99.7%** | **96.0%** | **86.2%** | **76.2%** | **66.0%** |

## K.1 CLEAN ACCURACY OF DIFFERENT SUBSET

We demonstrate clean accuracy obtained by models trained by different model diversity promotion methods with different selection configurations in Table 31.

Table 31: Clean accuracy achieved by different defended models employing diversity-promotion techniques on different datasets with a different random number of models.

| | | MNIST | | | |
|---|---|---|---|---|---|
| Quantity | Random | Disco(Ensemble) | Disco(DivDis) | Disco(DivReg) | Disco(SVGD+) |
| | 1 | 99.5% | 99.5% | 97.5% | 99.4% |
| 10 | 3 | 99.6% | 99.6% | 99.2% | 99.6% |
| | 5 | 99.7% | 99.6% | 99.4% | 99.6% |
| | 8 | 99.6% | 99.6% | 99.5% | 99.6% |
| | 1 | 99.5% | 99.5% | 97.6% | 99.0% |
| 20 | 3 | 99.6% | 99.6% | 99.3% | 99.5% |
| | 5 | 99.6% | 99.6% | 99.4% | 99.5% |
| | 10 | 99.7% | 99.6% | 99.6% | 99.6% |
| | 1 | 99.3% | 99.5% | 98.6% | 98.7% |
| | 3 | 99.5% | 99.5% | 99.4% | 99.2% |
| 40 | 5 | 99.5% | 99.6% | 99.5% | 99.4% |
| | 20 | 99.6% | 99.7% | 99.6% | 99.6% |
| | 30 | 99.6% | 99.7% | 99.6% | 99.6% |
| | | CIFAR-10 | | | |
| Quantity | Random | Disco(Ensemble) | Disco(DivDis) | Disco(DivReg) | Disco(SVGD+) |
| | 1 | 92.2% | 90.5% | 91.8% | 87.9% |
| 10 | 3 | 93.8% | 92.5% | 93.9% | 91.1% |
| | 5 | 94.0% | 93.3% | 94.3% | 92.3% |
| | 8 | 94.4% | 93.5% | 94.5% | 92.5% |
| | | STL-10 | | | |
| Quantity | Random | Disco(Ensemble) | Disco(DivDis) | Disco(DivReg) | Disco(SVGD+) |
| 10 | 5 | 91.6% | 90.2% | 89.7% | 88.2% |

## L ROBUSTNESS OVER MULTIPLE TRIALS

In this section, we conduct an extensive experiment to study the robustness of different defense mechanisms with randomness involvement. We evaluate RND, RF, and our proposed method against SQUAREATTACK ($l_2$) on an evaluation set of 500 correctly classified images drawn from CIFAR-10. Each defense is evaluated five times with different random seeds. Figure 5 presents the mean accuracy under attacks, with the upper and lower error bars representing the mean $\pm$ standard deviation. The results in Figure 5 show that the variation of our method is similar to other defenses and our lower error bar is far higher than the upper error bar of both RND and RF.

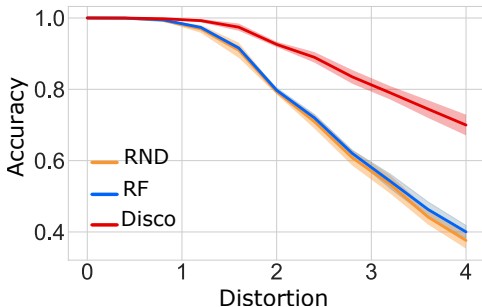

Figure 5: A comparison of the average robustness between different defenses against SQUAREAT-TACK ($l_2$). Mean accuracy under attacks, with the upper and lower error bars representing the mean $\pm 1\sigma$ (standard deviation).

## M  COMPARISON WITH ADVERSARIAL TRAINING (AT)

We conduct an experiment to demonstrate the robustness of our proposed method Disco, RND, RF and a state-of-the-art adversarial training (AT) (Wang et al., 2023) used for the CIFAR-10 task. We used the strong, query-based black-box attack, SQUAREATTACK under the $l_2$ objective. We use the implementation and the pre-trained model ($l_2$) from *Robustbench*[3] (Croce et al., 2021).

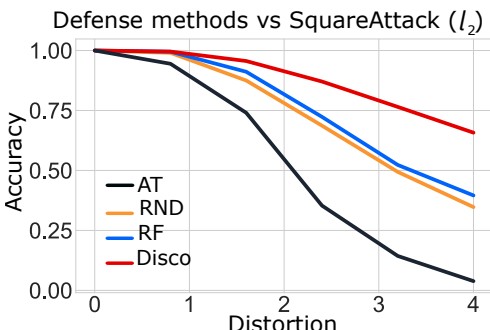

Figure 6: A comparison between Disco, RND, RF and AT against SQUAREATTACK ($l_2$).

The results in Figure 6, demonstrate that our simpler approach employing model radomization is better than the state-of-the-art adversarial training for a query-based black-box attack. Notably, our result comparison with AT also confirms those found in the recent black-box defense, RF Nguyen et al. (2024) where the AT methods itself was not as robust at the dedicated black-box defense (see AT vs. Ours in Table 4 in Nguyen et al. (2024)).

Notably, it should be mentioned here that the model randomization methods can also be adopted with adversarial trained models. We expect the robustness afforded by AT methods to then further improve the robustness from our approach. However, employing AT methods are likely to come at the cost of noticeable clean accuracy drops.

## N  COMPARISON WITH ADVERSARIAL ATTACK ON ATTACKERS (AAA)

AAA is a defense algorithm mainly designed for *score-based* attacks so it is expected to be successful against these attacks. Notably, it does not strongly withstand decision-based attacks as reported in evaluations by Nguyen et al. (2024). Nevertheless, we conduct an experiment to demonstrate the robustness of our proposed method versus AAA (Chen et al., 2022) on CIFAR-10 against the strong query-based black-box SQUAREATTACK ($l_2$) under the score-based setting. The results in Figure 7 demonstrate that our proposed defense mechanism is much more robust than AAA, especially at high perturbation budgets.

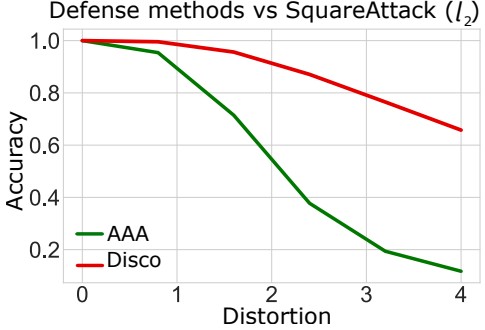

Figure 7: A robustness comparison between AAA and our method against SQUAREATTACK ($l_2$).

---

[3]https://github.com/RobustBench/robustbench

Table 32: Robustness comparision (higer is better) between our approach and adding noise to the output score under SQUAREATTACK ($l_2$, $l_\infty$)

| Methods | SQUAREATTACK ($l_2$) | | | | | SQUAREATTACK ($l_\infty$) | | | | |
|---|---|---|---|---|---|---|---|---|---|---|
| | $l_2 =0.8$ | 1.6 | 2.4 | 3.2 | 4.0 | $l_\infty=0.2$ | 0.4 | 0.6 | 0.8 | 1.0 |
| Add noise to output scores | 96.31 % | 78.61% | 62.64% | 41.61% | 31.97% | 98.09% | 80.64% | 52.08% | 29.1% | 20.28% |
| Disco | 99.56% | 95.62% | 87.07% | 76.5% | 65.76% | 99.97% | 96.91% | 86.52% | 70.22% | 55.77% |

Table 33: Robustness comparison (higher is better) between our approach and using two strategies to add noise to the output score under SquareAttack ($l_2$)

| Methods | $l_2 =0.8$ | 1.6 | 2.4 | 3.2 | 4.0 |
|---|---|---|---|---|---|
| Add constrained noise to output scores | 89.72 % | 75.13% | 46.58 % | 24.04 % | 16.89% |
| Disco | 99.56% | 95.62% | 87.07% | 76.5% | 65.76% |

## O  COMPARISON WITH THE MECHANISM OF ADDING NOISE TO THE OUTPUT SCORES

The difference between our approach for output diversity and simply adding noise to output scores is interesting to consider:

**Unconstrained Noise.** Based on our Proposition 3.2, it is true that the high diversity in output scores will result in a high probability of misleading gradient-free attacks. However, as we have seen with other random-noise based defences, this will come at the cost of model utility. To obtain high diversity in output scores, if we add large noise to the output scores, it can lead to unpredictable distortions and create unintended consequences due to its random nature. For instance, noise can cause the ground-truth class to lose its highest score ranking.

Instead, our approach aims to train an ensemble of models such that their predictions are consistent while their output scores are diverse, as mentioned in Section 3.6. We obtain this by learning a posterior distribution of model parameters. The underlying models' parameters sampled from that posterior distribution can result in diverse representations (Doan et al., 2022). This leads to output score variance while these underlying models mostly predict the same label for the same clean input and input samples in their neighbour regions. Thus, they maintain high clean accuracy or model utility. This demonstrates a key distinction between our approach and simply adding noise-based methods, in general. Perhaps, most importantly, the empirical evidence we have shown that the idea of model randomisation can outperform current noise-based methods. Additionally, the results in Tables 32 show that simply adding noise is much less robust than our approach.

**Constrained Noise.** we conducted a new experiment to explore the robustness of adding constrained noise to output scores based on the study in (Wen et al., 2021). We use the same noise strength as we did to collect the results in Table 32. Our new results in Table 33 demonstrate that the constrained noise injection is worse than the naive approach (unconstrained noise). This constraint might only work well for clean inputs (benigns). We highly suspect that, for inputs near the decision boundary (such as those crafted by adversaries with perturbations at some intermediate steps), sufficiently large perturbations in the next steps could push those inputs further and cross the boundary. To this end, the top-1 predicted label of the perturbed input (malicious) is already misclassified. Therefore, the constrained noise as in (Wen et al., 2021) is not necessarily better than the unconstrained noise and does not offer a clear advantage over unconstrained noise, particularly in adversarial attack settings.

## P  COMPARISON WITH REGION-BASED CLASSIFICATION (RBC)

Region-Based Classification (RBC) method (Cao & Gong, 2017) is a defense initially designed and evaluated with white-box attacks. It aims to add noise to the input and employ majority voting to make decisions. Given its strategy of adding random-noise to inputs, it can also be employed with black-box attacks. Therefore, in this section, we conduct extensive experiments to compare the robustness of our defense and the RBC method against score-based adversarial attacks NESATTACK ($l_\infty$), SIGNHUNTER ($l_2$, $l_\infty$), SQUAREATTACK ($l_2$, $l_\infty$), SPARSERS ($l_0$) and decision-based attacks

HOPSKIPJUMP ($l_2$) and SPAEVO ($l_0$) on `CIFAR-10`. The results in tables 34, 35, 36 and 37 show that our method is more robust than RBC.

Table 34: $l_2$ **objective attacks, `CIFAR-10`.** Robustness (higher ↑ is stronger) of different defense methods against SIGNHUNTER and SQUAREATTACK.

| Methods | SIGNHUNTER | | | | | SQUAREATTACK | | | | |
|---|---|---|---|---|---|---|---|---|---|---|
| | $l_2$ =0.8 | 1.6 | 2.4 | 3.2 | 4.0 | $l_2$ =0.8 | 1.6 | 2.4 | 3.2 | 4.0 |
| RBC | 25.86% | 10.34% | 3.55% | 1.79% | 1.18% | 8.64% | 2.65 % | 0.95% | 0.28% | 0.05% |
| **Disco** | **99.96%** | **99.25%** | **97.61%** | **93.63%** | **90.24%** | **99.56%** | **95.62%** | **87.07%** | **76.5%** | **65.76%** |

Table 35: $l_\infty$ **objective attacks, `CIFAR-10`.** Robustness (higher ↑ is stronger) of different defense methods against attacks NESATTACK, SIGNHUNTER and SQUAREATTACK.

| Attack | Methods | $l_\infty$ =0.02 | 0.04 | 0.06 | 0.08 | 0.1 |
|---|---|---|---|---|---|---|
| NESATTACK | RBC | 94.06% | 86.07 % | 79.08% | 76.03% | 73.34% |
| | **Disco** | 99.7% | 97.93% | 94.39% | 90.5% | 86.77% |
| SIGNHUNTER | RBC | 29.69% | 6.99 % | 3.75% | 0.93% | 0.53% |
| | **Disco** | 99.97% | **98.95%** | **95.56%** | **90.7%** | **84.22%** |
| SQUAREATTACK | RBC | 37.4% | 6.32 % | 2.63% | 0.1% | 0.1% |
| | **Disco** | 99.97% | **96.91%** | **86.52%** | **70.22%** | **55.77%** |

Table 36: $l_0$ **objective attacks, `CIFAR-10`.** Robustness (higher ↑ is stronger) of different model diversity promotion schemes against SPARSERS.

| Methods | $l_0$ =16px | 32px | 48px | 64px | 80px |
|---|---|---|---|---|---|
| RBC | 0.43% | 0.29% | 0.2% | 0.17% | 0.01% |
| **Disco** | **63.85%** | **47.84%** | **41.59%** | **36.81%** | **31.24%** |

Table 37: **Decision-based.** Robustness (higher ↑ is stronger) of different model diversity promotion schemes against HOPSKIPJUMP ($l_2$) and SPAEVO ($l_0$) on the `CIFAR-10` task.

| Methods | HOPSKIPJUMP | | | | | SPAEVO | | | | |
|---|---|---|---|---|---|---|---|---|---|---|
| | $l_2$ =0.8 | 1.6 | 2.4 | 3.2 | 4.0 | $l_0$ =4px | 8px | 12px | 16px | 20px |
| RBC | 94.44% | 79.24 % | 61.47% | 57.54% | 57.89% | 55.21% | 31.18% | 16.16% | 8.29% | 8.29% |
| **Disco** | **99.94%** | 99.4% | 98.77% | **98.04%** | **97.43%** | **96.17%** | **95.99%** | **95.94%** | **95.88%** | **95.84%** |

# Q FUNDAMENTAL DIFFERENCES FROM PURIDEFENSE

While both our work and Guo et al. (2024) leverage model randomization to enhance robustness against query-based black-box attacks, there are fundamental differences in methodology, and theoretical analysis that distinguish our approach:

**Methodology.** Puriefense in Guo et al. (2024) employs a random patch-wise purification mechanism that leverages the randomized local implicit functions to remove adversarial perturbations and rebuild the natural image manifold. Particularly, adversarial image patches are randomly purified by using the local implicit function, which is an ensemble of lightweight purification models before being classified by a fixed model. This approach primarily operates in the input space. In contrast, our work directly selects a random subset of models to introduce diversity at the model level. Instead of relying on input-space transformations, our defense mechanism disrupts gradient estimation by modifying the model architecture or weight distribution at inference time.

Additionally, while PuriDefense employs distinct architecture with an adversarial training loss, we focus on the same architecture with a training loss that aims to promote model diversity. We introduce a new learning objective to enhance model diversity, which is absent in Guo et al. (2024). Moreover, PuriDefense requires adversarial examples generated by different white-box attacks to train purification models, whereas we do not require synthetic data.

**Theoretical Analysis.** The theoretical foundation of PuriDefense focuses on the efficacy of employing randomized local implicit functions to purify adversarial perturbation and its impact on query-based attack success rates. Their analysis primarily examines how randomized purification reduces adversarial vulnerability. Our work, however, provides a formal analysis of how model-level

randomization affects the convergence and reliability of gradient-based query attacks, demonstrating how it fundamentally disrupts query-based optimization methods. Furthermore, PuriDefense does not possess the capability of directly promoting diversity as in Guo et al. (2024); the diversity of the purifiers is associated the diversity with the number of purification models. The larger the number of purifiers, the more diversity is. In contrast, our approach purposely introduces controllable diversity by directly pushing model parameters apart through SVGD framework. Therefore, the impact of our approach on the convergence rate is more guaranteed.

## R  DISCUSSION, LIMITATION AND BROADER IMPACT

**Discussion of A Possible New Adaptive Attack Leveraging ILP-based Gradient Recovery**

We sincerely thank the reviewer for the detailed and thoughtful suggestion regarding a possible adaptive attack leveraging ILP-based gradient recovery. The proposed approach is both creative and technically rich, and we appreciate the clear exposition of its underlying assumptions and potential applicability. While implementing this attack in full would require non-trivial engineering effort and exploration of solver capabilities, we agree that it represents an interesting and promising direction for evaluating the limits of our defense. Thus, we incorporate the reviewer's idea into our comprehensive analysis, explicitly discussing its feasibility, potential impact, and the scenarios under which it might succeed or fail. We also see significant value in this concept for informing future work—both in designing stronger adaptive attacks and in building more robust defenses. This contribution will thus not only enhance the depth of our current paper but also serve as a useful reference point for the research community.

**Technical Discussion.** Regarding the adaptive attack discussed in Appendix A.3 it can be generalized to $N > 1$. For instance, consider the following approach using integer linear programming. Submit the input $x$, $M$ times, and record all $M$ responses. Also submit the input $x + \epsilon u$, $M$ times. Introduce $2CN$ continuous variables $v_{i,c}$ and $g_{i,c}$ (where $C$ is the number of classes, i.e., the number of logit outputs, and $i \in [N]$ identifies a model and $c \in [C]$ identifies a class), and $2NM$ 0-or-1 variables $y_{i,j}, z_{i,j}$. The intention is that $v_{i,c}$ represents the logit output for class $c$ from the $i$th model when provided $x$ as input, and $g_{i,c}$ represents the "gradient", more precisely, $v_{i,c} + \epsilon g_{i,c}$ is the logit output for class $c$ from the $i$th model on input $x + \epsilon u$. Also the intention is that $y_{i,j} = 1$ if the $i$th model was one of $N$ selected when computing the $j$th response to input $x$, and similarly for $z_{i,j}$ except for input $x + \epsilon u$. We'll use an ILP solver to try to solve for the $v, g$ values. Add inequalities to enforce $v_{i,j,c} = v_{i,c} * y_{i,j}$; and enforce $g_{i,j,c} = g_{i,c} * z_{i,j}$. Assuming that logit outputs are in the range $[0, 1]$, and made a heuristic assumption that the Linf norm of all gradients is at most 10.) Add the linear equalities $\sum_i y_{i,j} = \sum_i z_{i,j} = N$. Finally add a linear inequality to enforce that $\sum_i v_{i,j,c}$ is close to the observed logit for class $c$ in the $j$th repeat, and similarly for $\sum_i g_{i,j,c}$. We can use off-the-shelf ILP solver to find a feasible solution to this ILP.

How do you enforce $a = b * c$, in ILP? Let's assume that $a, b$ are in the range $[-20, 20]$ (probably reasonable for both logits and gradients, as a heuristic). Write $a = a^+ - a^-$, $0 \le a^+, a^- \le 20$, and similarly for $b$, and then enforce $a^+ = b^+ * c$ and $a^- = b^- * c$, which reduce to the case where $a, b$ are in $[0, 20]$. In that case you can enforce $a^+ = b^+ * c$ by adding linear inequalities $0 \le a^+ \le b^+$, $b^+ - 20(1 - c) \le a^+ \le 20c$ and similarly for $a^-$. If the ILP solver finds a solution, then you have an inferred/assumed gradient for each of the K models. Then you can take a gradient update step knowing all of this information, e.g., moving in the direction $\sum_i g_{i,c}$.

Will this work to find the gradients? The answer may not be clear but it might depend on how much information is available and what off-the-shelf ILP solvers can do. If we want to test it, we might start by trying it with the case $N = 2$, $K = 5$, as used for `ImageNet`. If the softmax is output in 16 bits of precision, and $N = 2$, $K = 10$, $C = 10$, and all the softmax values are different enough, then maybe with about $M = 5$ or $M = 10$ repeats, there might be enough information to recover most of the model outputs, and perhaps an ILP solver could find it.

Will finding all of the model gradients be enough to break the scheme? Although the answer is unclear, it seems like if an attacker can find all the model gradients, then the argument for the security of the scheme looks rather thin. However, our results and discussion in Appendix A.3 show that it is not the case and this proposed adaptive attack is weaker than EOT style adaptive attack in the case of selecting $N = 1$ out of $K$ models.

**Limitation.** One key limitation of our approach is the computational and resource overhead they impose. For instance, training multiple models can be time-consuming, storing a diverse set of models can be memory-intensive and inference may require high latency. However, as outlined in Appendix E, we can mitigate this problem by leveraging LoRA method. As we are now focusing on classification models whose model size is moderate, the scalability of our method can be well addressed by employing LoRA. When moving to foundation models, it could be more challenging, even with LoRA. However, we leave it for our future work to explore the capability of our approach to defend foundation models against black-box attacks. This promising avenue holds the potential for significantly bolstering the robustness of machine learning models in practical applications.

**Broader Impact.** This study provides a substantial contribution to the field by introducing an efficient and effective defense mechanism that significantly mitigates the threat of existing query-based black-box adversarial attacks, with only a minimal reduction in clean accuracy, thereby ensuring its practical applicability. Additionally, our defense strategy has the potential to inspire other researchers to develop even more efficient and practical defense methods. Nonetheless, a potential negative societal impact of our work is the possibility that it may incentivize malicious actors to develop new attack methods once they become aware of our proposed mechanism.

