# OpenReview forum: "Black-box Attack Robustness with Model Diversity and Randomization"
_ICLR.cc/2026/Conference — Submitted to ICLR 2026_

### Official Review · Reviewer_YRWy · 2025-11-01

**Soundness:** 3
**Presentation:** 3
**Contribution:** 2
**Rating:** 2
**Confidence:** 4

**Summary:**

This paper proposes a novel defense mechanism called Disco (Diversity Induced Stochastic Obfuscation) against query-based black-box adversarial attacks. Unlike existing defenses that rely on adding random noise to inputs or features, the authors investigate randomizing model parameters by training a diverse set of well-performing models and randomly selecting a subset to respond to each query.

The key insight is that query-based attacks depend on successive queries with fixed model parameters to estimate gradients or search directions. By randomizing which models respond to each query, the defense obfuscates the relationship between query-response pairs, making it significantly harder for attackers to extract useful information.

**Strengths:**

1.  The paper provides comprehensive experimental evaluation over 7 state-of-the-art attacks across all three perturbation norms.

2. The paper provides formal analysis of the defense mechanism: Proposition 3.1 quantifies query complexity, Proposition 3.2 bounds the probability of misleading gradient-free attacks, with analysis extends to adaptive attacks with EOT.

3. The paper addresses real-world deployment concerns: (1) LoRA Integration (2) Multiple Diversity Methods (3) Cost Analysis.

4. The method achieves consistently superior robustness across various settings.

**Weaknesses:**

1. **Fundamental Novelty Concerns Regarding PuriDefense [1]**

Both papers exploit the same fundamental principle: *randomizing which function processes each query disrupts the inter-query relationships that attacks rely on*. While the authors claim differences (Appendix Q), the conceptual framework is strikingly similar:
 - PuriDefense: Maintains K purification models, randomly selects N to process input patches, feeds result to fixed classifier
 - Disco: Maintains K full classification models, randomly selects N to generate response

They both everage the principle that random selection from a diverse pool obfuscates gradient estimation and search directions.
The distinction between "input-space" vs. "model-space" randomization appears to be an implementation detail rather than a conceptually novel defense paradigm.
From an attacker's perspective, both methods present a randomly varying function $f_i$ sampled from a set ${f_1, ..., f_K}$.

2. **Theoretical Contribution Overlap**

The theoretical analysis appears to formalize what PuriDefense [1] already demonstrated:

 - Proposition 3.1: More queries needed when models are diverse -- This is intuitive and was shown empirically/theoretically in [1]
 - Proposition 3.2: Higher probability of misleading gradient-free attacks with diversity -- Again, this follows directly from the randomization principle established in [1]

The mathematical formalization using Hoeffding's inequality (Eq. 7) and Markov/Jensen inequalities (Eq. 8) provides rigor but doesn't offer fundamentally new insights beyond "diversity increases query complexity," which [1] already established.

3. **Incremental Nature of Contribution**:

Given PuriDefense established the model randomization paradigm, this work appears incremental:
- **Training Method (SVGD+)**: The sample loss addition is a reasonable engineering improvement but doesn't constitute a major conceptual advance
- **Diversity Promotion**: Multiple prior works (cited: DivDis, DivReg, ADP) already explored diversity for robustness
- **Application**: Applying existing diversity techniques within the PuriDefense framework seems evolutionary rather than revolutionary

[1] Guo et al. (2024), "Puridefense: Randomized local implicit adversarial purification for defending black-box query-based attacks" (arXiv:2401.10586).

**Questions:**

Q1: From a defense mechanism perspective, both PuriDefense and Disco implement the same high-level strategy: maintain $K$ diverse functions, randomly sample $N$ for each query, and use their combined output. Can the authors articulate why moving the randomization from input-space purification (PuriDefense) to model-space selection (Disco) constitutes a fundamentally novel defense paradigm rather than an implementation variant?

Q2: Proposition 3.1 essentially states: "diverse models require more queries for gradient estimation." Proposition 3.2 states: "diverse models increase probability of misleading gradient-free attacks."
Doesn't PuriDefense already establish these principles (theoretically and empirically) for their randomized purifiers? What fundamentally new theoretical insight do these propositions provide beyond formalizing the query complexity of a randomization-based defense that PuriDefense already demonstrated?
Could the authors identify specific theoretical results that would be:
- (a) Impossible to derive from PuriDefense's framework?
- (b) Provide actionable insights not available from PuriDefense?

---

### Official Review · Reviewer_gxrF · 2025-11-02

**Soundness:** 1
**Presentation:** 2
**Contribution:** 1
**Rating:** 2
**Confidence:** 5

**Summary:**

The paper's goal is to design is to design a defense against adversarial examples in the black-box setting. The idea behind the proposed defense is to sample from a distribution of models, and serve incoming queries with such a sampled model. The model distribution is constructed such that model parameters satisfy some diversity property, with the intuition that the resulting classifier is no longer smooth, making it harder to estimate a quality gradient with fewer points.

**Strengths:**

- It is commendable to tackle image-based adversarial examples, given progress in recent years.
- The presentation of the paper is neat.

**Weaknesses:**

The primary weakness of this work is that this style of defense has already been studied, and is known to not work.

To elaborate, the adversarial machine-learning literature has converged upon several lessons in the past few years, with one being that defenses based on randomness *typically* do not work. As an aside, they also tend to make evaluation more confusing, but that is not the concern here. In this case, the specific class of defenses based on random transformations/ensemble diversity are very well known to lack any kind of meaningful robustness properties when a perturbation is (properly) computed in expectation across the randomness. Yes, more queries are needed to estimate a gradient when the function is not smooth. The defense still breaks, and as a community we are already well aware that it will take a few more queries to do this. You can see this is your results on the EoT "adaptive" attack, where robustness goes down. I am quite confident that if I or someone else could further worsen this curve with stronger implementations of EoT, or with even more queries. Note that adaptive attacks are the more important evaluation criterion when concerned with adversarial examples, i.e., we are truly concerned with worst-case behavior here. In fact, zero-knowledge adversaries are practically unlikely for image classification, particularly in today's times with the vast body of work on how to break defenses.

I would like to communicate some other points. Note that you are considering a gradient of a gaussian smoothed version of your classification function, not the function itself. That is the premise of evolutionary strategies for black-box optimization. Here, the analysis is somewhat superfluous, i.e., the takeaway from the first result is a somewhat known concept - the variance of your estimate of the search distribution gradient goes down with more query samples. This comes from basic probability, and there are tons of results on this already. For example, consider a silly adversary, that makes their own life harder by using a gaussian with a very large \sigma - this is in a similar spirit, and you can find results on the quality of the estimated gradient in terms of \sigma. Less concerningly, but still a bit odd - with the way it is presented, I am not sure why one should concern themself with the *difference* between gradient estimate and true gradient - how that would that affect the optimization procedure?

I am sure it is possible to raise arguments about the computational difficulty of serving such a defense. Yes, it would be annoying to rotate models. But this is not *my* concern - my problems with the work begin and end with the already-known robustness properties. Also, I would recommend playing with more modern black-box attacks, like SurFree, QEBA, etc. There are so many newer ones, they are really quite efficient.

Overall, it is clear that effort has been put into writing and presenting the paper, and I appreciate that and respect the work from the authors towards trying to build a defense. This is a hard problem, but unfortunately, this is definitely not the solution.

**Questions:**

How is this fundamentally any different from a canonical randomization-based defense? One could randomize the input query, randomize the model served, randomize anything. This simply builds a distribution around the problem. What is special about this defense, such that one cannot optimize in expectation across the distribution?

---

### Official Review · Reviewer_DwGH · 2025-11-09

**Soundness:** 4
**Presentation:** 4
**Contribution:** 3
**Rating:** 8
**Confidence:** 2

**Summary:**

This paper proposes Disco, an adversarial defense that randomly samples a model from a diverse ensemble to answer each query, breaking the attacker’s assumption that the target model is fixed across queries. The authors perform theoretical and empirical analysis and demonstrate that this approach significantly improves robustness across multiple attack types and perturbation norms, while maintaining clean accuracy.

**Strengths:**

- clearly written paper
- extremely thorough experiments, comparison to baselines, exploration of the method
- theoretical analysis
- I especially appreciated the discussion of cost/practicality (Sec 4.5) -- the fact that Disco works with LoRA is an important point

**Weaknesses:**

- impracticality of the proposed method
	- need to train multiple models (10-40 per dataset) and do multiple forward passes at inference time
	- incompatible with deterministic APIs, which one often wants in practice
- weak threat model and security guarantees
	- method operates by preventing gradient-based attacks from converging through stochasticity in function space. However, prior work [1] has argued how methods like these do not fundamentally improve the security of the system
	- generally makes attacks more *difficult* without preventing them: e.g. only partial robustness to adaptive attacks like EOT. DISCO requires 10x query cost, but doesn't fundamentally change the robustness of the system.
	- easily detectable by an attacker (and perhaps more easily defeated by a corresponding attack method) by observing that outputs differ across calls

Given these weaknesses, I don't think this paper represents a fundamental advancement in adversarial defenses. However, I still believe this paper is a well-executed, strong contribution to the community's scientific understanding of the problem.

**Questions:**

- how do we reconcile the effectiveness of this defense with the fact that adversarial examples transfer across models? Would this be robust if the attacker optimizes against an ensemble of models? (I may have missed this experiment somewhere, but would generally be curious for the authors' thoughts on this)
- describe what "single (undef)" and "ensemble (undef)" are in Table 1

---

### Official Review · Reviewer_5i4w · 2025-11-10

**Soundness:** 3
**Presentation:** 3
**Contribution:** 3
**Rating:** 4
**Confidence:** 4

**Summary:**

This paper proposes a defense against query-based black box attacks. Unlike defenses that inject random noise into inputs or features, Disco randomizes model parameters by responding to each query using a random subset of models sampled from a diverse pool. Theoretical analysis is provided to show that increased model diversity degrades the attacker's gradient estimation. Extensive evaluation is performed on MNIST, CIFAR-10, STL-10, and ImageNet.

**Strengths:**

- valuable theoretical results
- extensive evaluation

**Weaknesses:**

- My main concern is regarding the notion of using obfuscated gradients as a defense. Past work has demonstrated again and again that any advantage offered by the defense is usually due to inefficiencies in the attacks (Obfuscated Gradients Give a False Sense of Security: Circumventing Defenses to Adversarial Examples, by athalye et al.). The authors should evaluate against stronger adaptive attacks such as AutoAttack and also analyze attack progression with increasing number of steps.

- The authors should also compare with other work on diversity based defenses such as TRS: Transferability Reduced Ensemble via
Promoting Gradient Diversity and Model Smoothness, by Yang et al.

**Questions:**

NA

---

### Official Review · Reviewer_sojJ · 2025-11-12

**Soundness:** 4
**Presentation:** 3
**Contribution:** 3
**Rating:** 2
**Confidence:** 4

**Summary:**

They propose a defense against black-box adversarial attacks. The main idea is to train multiple models, randomly select some of them, and output the mean logit.

**Strengths:**

1) With theoretical analysis.
2) The training loss and efficiency of multiple models are improved.
3) The experimental results demonstrate superiority.

**Weaknesses:**

1) I find a counter-example against Proposition 3.1. Let the probability mass function of $g(x)$ be
$$
\begin{cases}
\epsilon,&g(x)=a\\\\
1-2\epsilon,&g(x)=0\\\\
\epsilon,&g(x)=b
\end{cases}
$$
where $a<0<b$ and $0<\epsilon<0.5$. Then we have $a\leq g(x)\leq b$. Let $|a-b|\to\infty$ and then the bound in (7) can be arbitrarily large. Meanwhile, let $\epsilon\to 0$ and then $g(x)$ can be estimated with arbitrary precision, that is, $n$ can be small. It is a contradiction.

2) Proposition 3.2 merely indicates the vulnerability of low diversity, rather than the robustness of high diversity. You should provide the lower bound instead of upper bound.

3) The proof employs Taylor approximation, which may lead to errors.

Minor revisions: At Lines 771~777, $\pmb{u}\nabla F(\pmb{x})$ should be $\pmb{u}^T\nabla F(\pmb{x})$, and the last two $\approx$ should be $=$. Similar mistakes occur repeatedly in other places.

**Questions:**

What are the values of $\mu_1,\mu_2,\ldots,\mu_K$ in your experiment?

---

### Meta-Review · Area_Chair_VDJY · 2026-01-05

**Summary:**

This submission proposes a defense method against query-based black-box adversarial attacks that relies on random sampling from a diverse model ensemble to obfuscate the relationship between successive query-response pairs. Notably, the authors did not submit any rebuttal to address these concerns during the discussion phase.

**Reviewer Concerns:**

All key concerns raised by reviewers remain outstanding, as the authors failed to provide any rebuttal or supplementary explanation.

**Reviewer Scores:**

Given the authors' lack of rebuttal to address any concerns, the reviewers' scores are expected to remain consistent with their original ratings.

---

### Decision · Program_Chairs · 2026-01-26

Reject